# DAF-16/FoxO and DAF-12/VDR control cellular plasticity both cell-autonomously and via interorgan signaling

Ulkar Aghayeva[1], Abhishek Bhattacharya[1], Surojit Sural[1], Eliza Jaeger[1], Matthew Churgin[2], Christopher Fang-Yen[2], Oliver Hobert[1]*

**1** Department of Biological Sciences, Howard Hughes Medical Institute, Columbia University, New York, New York, United States of America, **2** Department of Bioengineering, School of Engineering and Applied Science, University of Pennsylvania, Philadelphia, Pennsylvania, United States of America

☯ These authors contributed equally to this work.
* or38@columbia.edu

**Data Availability Statement:** All relevant data are within the paper and its Supporting Information files.

## Abstract

Many cell types display the remarkable ability to alter their cellular phenotype in response to specific external or internal signals. Such phenotypic plasticity is apparent in the nematode *Caenorhabditis elegans* when adverse environmental conditions trigger entry into the dauer diapause stage. This entry is accompanied by structural, molecular, and functional remodeling of a number of distinct tissue types of the animal, including its nervous system. The transcription factor (TF) effectors of 3 different hormonal signaling systems, the insulin-responsive DAF-16/FoxO TF, the TGFβ-responsive DAF-3/SMAD TF, and the steroid nuclear hormone receptor, DAF-12/VDR, a homolog of the vitamin D receptor (VDR), were previously shown to be required for entering the dauer arrest stage, but their cellular and temporal focus of action for the underlying cellular remodeling processes remained incompletely understood. Through the generation of conditional alleles that allowed us to spatially and temporally control gene activity, we show here that all 3 TFs are not only required to initiate tissue remodeling upon entry into the dauer stage, as shown before, but are also continuously required to maintain the remodeled state. We show that DAF-3/SMAD is required in sensory neurons to promote and then maintain animal-wide tissue remodeling events. In contrast, DAF-16/FoxO or DAF-12/VDR act cell-autonomously to control anatomical, molecular, and behavioral remodeling events in specific cell types. Intriguingly, we also uncover non-cell autonomous function of DAF-16/FoxO and DAF-12/VDR in nervous system remodeling, indicating the presence of several insulin-dependent interorgan signaling axes. Our findings provide novel perspectives into how hormonal systems control tissue remodeling.

## Introduction

The identity of a fully differentiated cell in a multicellular organism is usually described by a number of phenotypic criteria, ranging from overall anatomy to cellular function to molecular

**Funding:** This work was supported by NIH R21NS115442 (OH) and NIH R01NS-084835 (C. F-Y.). OH is an Investigator of the Howard Hughes Medical Institute. The funders had no role in study design, data collection and analysis, decision to publish, or preparation of the manuscript.

**Competing interests:** The authors have declared that no competing interests exist.

**Abbreviations:** AID, auxin-inducible degron; DA, dafachronic acid; IAA, indole-3-acetic acid; NGM, Nematode Growth Medium; RF, restriction-free; rGC, receptor type guanylyl cyclase; TF, transcription factor; VDR, vitamin D receptor.

features. Once such differentiated state has been acquired, it is often thought to persist throughout the life of an animal and be controlled by active maintenance mechanisms [1]. Nonetheless, Rudolf Virchow already pointed out in 1886 that there are "plastic processes" that accompany the transition of a differentiated cell into a different state, particularly in the context of disease [2]. The plasticity of cellular phenotypes has now become a widely accepted phenomenon, occurring in many different cellular and organismal contexts [2]. In the brain, cellular plasticity phenomena become evident in a number of distinct contexts and include cellular remodeling events that occur after injury, after the encounter of stressful environmental conditions, or during specific developmental transitions, such as puberty. Several hormonal signaling systems have been implicated in triggering these structural and functional remodeling events in the vertebrate brain [3].

Remarkable changes of cellular phenotypes within distinct tissue types, including the nervous system, are observed upon entry in the dauer stage, a diapause stage of rhabditid nematodes [4]. In response to detrimental environmental conditions perceived at a specific postembryonic larval stage, epidermal and muscle cell types shrink their volume, resulting in an overall constriction of animal shape [4–6]. The extracellular collagen cuticle, secreted by skin cells, is remodeled, and intestinal cells alter their metabolic state [5,7]. Several profound cellular remodeling events are observed in the nervous system: Some sensory neurons grow extensive dendritic branches [8], some glial cells change their ensheathment of several neuron types [9,10], and there is extensive remodeling of electrical synapses throughout the entire nervous system [11]. The expression patterns of many G-protein–coupled sensory receptors are altered in a highly cell type–specific manner in a number of sensory, inter-, and motor neuron types [12–14]. On an animal-wide level, there are global changes in gene expression and chromatin modification patterns [15], including widespread changes in neuropeptide gene expression [16]. Paralleling these changes in cellular and molecular phenotypes, animals undergo a number of striking behavioral changes, from altered sensory responses to changes in locomotory and exploratory strategies to a complete silencing of the enteric nervous system [4,11,17,18]. Most of these remodeling events are reversible. After the improvement of external conditions, animals exit the dauer stage and develop into adults whose anatomy and behavior at least superficially resembles that of worms that have not passed through the dauer stage, even though some molecular changes appear to persist [14,15,19].

Most cells that are remodeled upon entry into the dauer state have already been born and differentiated into fully functional cell types during embryogenesis and early larval development. The ability of postmitotic, differentiated cells to remodel a number of their anatomical, molecular, and functional features during dauer entry in mid-larval development (and their ensuing reversibility upon dauer exit) is a remarkable example of the plasticity of the cellular phenotype. How are these remodeling events superimposed onto the regulatory state of a differentiated neuron? Previous screens for mutants with an impaired dauer entry phenotype (Daf-d for "dauer defective" or Daf-c for "dauer constitutive") have identified 3 hormonal systems involved in this process: an insulin-like signaling pathway, a TGFβ signaling pathway, and a nuclear steroid receptor pathway (**Fig 1A**) [20–25]. While these pathways have been studied to a varying extent in different tissue types, it remains to be better understood where and when these systems act to remodel individual tissue types, specifically in regard to the remodeling of the nervous system.

The focus of action of the TGFβ signaling pathway for dauer formation appears to lie in the nervous system, as assessed by the mosaic analysis of the *daf-4* TGFβ receptor [26] and the rescue of neuronal remodeling events through neuronal expression of *daf-4* [13]. The situation is less clear for the insulin-like receptor pathway, whose major effector is the ubiquitously expressed DAF-16/FoxO transcription factor (TF), which translocates into the nucleus upon

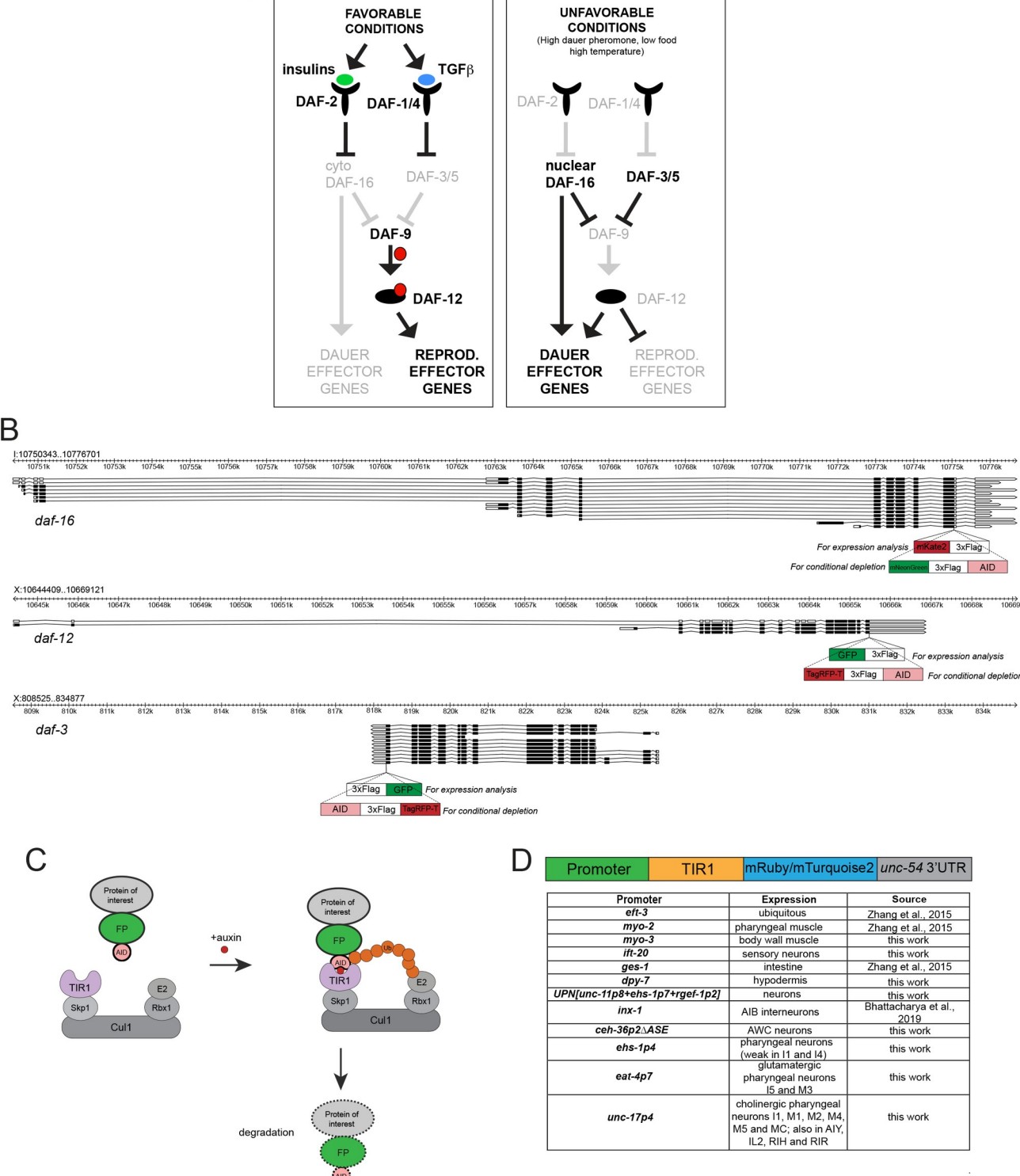

**Fig 1. Reagents generated for this study.** (**A**) Much simplified overview of the dauer pathways, with DAF-3/SMAD, DAF-16/FoxO, and DAF-12/VDR as transcriptional effectors [25]. (**B**) Genomic loci of *daf-16*, *daf-12*, and *daf-3*. The insertions sites and schematics of the expression reporters and of the AID tag are shown. (**C**) Schematic of the AID system. Skp1, Cul1, Rbx1, and E2 are phylogenetically conserved components of the E3 ligase complex. TIR1 is a plant-specific substrate-recognizing subunit of the E3 ligase complex. AID, fused to a protein of interest, is bound by TIR1 in the presence of auxin, which leads to ubiquitination and proteasomal degradation of the protein of interest. (**D**) Summary of TIR1 transgenes utilized in this study. See Materials and methods for details on TIR constructs [11,37–40]. AID, auxin-inducible degron.

entry into the dauer stage [27]. Two studies, using transgenic rescue approaches, have proposed distinct foci of action for *daf-16* in controlling entry into the dauer stage: One study showed that supplying *daf-16* into the nervous system of *daf-16*; *daf-2* double mutant animals restored their ability to undergo dauer remodeling [28], while another study showed that the rescue is only achieved with intestinal expression of *daf-16* [29].

The focus of action of DAF-12 for dauer remodeling events is even less clear [19,30]. DAF-12 is a nuclear steroid receptor that responds to *Caenorhabditis elegans* steroids, the dafachronic acids (DAs), which represent endogenous signals of the feeding state of the animal [31]. DAF-12 and its paralogue NHR-8 are similar to the vertebrate xenobiotic nuclear receptors PXR and CAR and to the VDR protein, a nuclear receptor for vitamin D, an endogenously produced neurosteroid [32]. Intestinally expressed NHR-8, a likely PXR and CAR ortholog, responds to xenobiotic substances, while DAF-12, like the vertebrate VDR, responds to endogenously produced steroids [33,34]. Similar to DAF-12, the vertebrate VDR is broadly expressed in the brain and mediates the many effects that vitamin D has on neuronal development and plasticity [32]. The focus of action of VDR is not well understood, and the same holds for DAF-12, as far as any tissue remodeling event during dauer entry is concerned.

In this paper, we address a number of presently unresolved questions about these 3 signaling systems. Do they operate autonomously within target tissues that undergo cellular remodeling? Or do they serve to relay internal signals to nonautonomously control cellular remodeling? Do they operate together within specific cellular context, or do they work sequentially? Do they only control the initial remodeling events upon dauer entry, or are they continuously required to maintain the remodeled state? To address these questions, we generated conditional and fluorescently labeled alleles of the effectors of each of the 3 hormonal signaling systems, the SMAD TF DAF-3, the FoxO TF DAF-16, and the steroid receptor DAF-12. We show that each of the 3 TFs displays distinctive dynamics in expression and localization throughout different cell types during continuous development and dauer entry. Temporally controlled removal shows their importance not only in initiating but also maintaining the remodeled state. Tissue- and cell type–specific removal of each of these TFs reveals complex requirements for these systems and shows that one hormonal axis acts mostly cell-autonomously, while two others act in both target cells (i.e., cell-autonomously), but also act outside the remodeled organ. Such cell nonautonomous activities reveal a number of interorgan signaling axes, from the nervous system to the gut and muscle and also from the gut to the nervous system. We show that in the context of the nervous system these hormonal signaling systems cooperate with hardwired terminal selector-type TFs that provide the cellular specificity and potential for remodeling.

## Results

### Expression pattern of daf-3/SMAD, daf-12/VDR, and daf-16/FoxO

The expression patterns of *daf-3/SMAD*, *daf-16/FoxO*, and *daf-12/VDR*, the transcriptional effectors of the 3 hormonal signaling systems controlling diapause (**Fig 1A**), have previously been examined with multicopy reporter transgenes [23,24,35,36]. To avoid potential issues with reporter transgenes (such as overexpression or lack of *cis*-regulatory control elements), we tagged the endogenous *daf-3/SMAD*, *daf-16/FoxO*, and *daf-12/VDR* loci with different fluorescent tags using CRISPR/Cas-9–mediated genome engineering (**Fig 1B**). For all 3 genes, we chose carboxyl-terminal fusions, since they tag all isoforms of the respective loci (**Fig 1B**). The analysis of the 3 endogenously tagged reporter alleles largely corroborated previously described sites of expression, based on reporter constructs, but also revealed novel spatiotemporal dynamics of gene expression.

The *daf-3* reporter alleles *ot875* and *ot877* generally have very low expression levels (**Fig 2A**). The expression is most prominent in L1 larvae, where it is observed in all tissue types, whereas it is down-regulated in subsequent larval stages and in adults, being only detectable in neurons and the intestine. Starvation up-regulates *daf-3*::*gfp* levels not only in L1 larvae, but also in later larval stages and adults (**Fig 2A**, **S2 Fig**). In dauers, expression levels of *daf-3*::*gfp* are lowest, with only nuclei of some head neurons visible.

In well-fed larvae and adults, the *daf-16* CRISPR reporter alleles *ot821* and *ot853* are expressed in the cytoplasm of neurons, most prominently, in sensory neurons of the head (**Fig 2B**). Nonneuronal tissues show only weak expression. After food deprivation, the tagged DAF-16/FoxO protein is up-regulated in nonneuronal tissues, where it is mostly nuclear (**Fig 2B**). However, in neurons of food-deprived larvae and adults, DAF-16 remains mostly cytoplasmic. In striking contrast, in dauers, the tagged DAF-16 protein displays nuclear localization throughout all tissues, including neurons (**Fig 2B**).

The *daf-12* reporter alleles *ot870* and *ot874* displayed ubiquitous expression, with highest expression levels during reproductive development at the L2 stage (**Fig 2C**), largely correlating with the timing of *daf-12's* heterochronic gene activity [62]. While starvation at different stages does not appear to impinge on its expression level, *daf-12/VDR* expression is significantly stronger throughout all tissues in the dauer stage (**Fig 2C**), contrasting earlier reports using a multicopy, cosmid-based reporter array of *daf-12* [24].

## DAF-3/SMAD, DAF-12/VDR, and DAF-16/FoxO are continuously required to maintain the dauer stage

To deplete gene function in a spatially and temporally controlled manner, we generated conditional alleles by inserting an auxin-inducible degron (AID) into each of the loci [41] (**Fig 1B and 1C**). This provided a genetic strategy that is orthogonal to the cell-specific rescue approaches previously used to assess the focus of *daf-16/FoxO* gene function [28,29,42]. For *daf-3/SMAD* and *daf-12/VDR*, no studies on the cellular focus of action for the dauer decision have been conducted so far.

We found that tagging each of the TF loci had no detectable effect on gene function in the context of dauer formation, as assessed by crossing the tagged loci into distinct Daf-c mutant backgrounds, whose Daf-c phenotype is known to be suppressed by reduction of *daf-3/SMAD*, *daf-12/VDR*, or *daf-16/FoxO* gene function (described further below). No suppression effects that would have been indicative of compromised gene function were observed with the tagged loci for any of the 3 TFs.

To achieve spatiotemporally controlled gene depletion, we generated transgenic lines that express the substrate-recognizing subunit of the plant ubiquitin ligase, TIR1, an essential component of the AID system [41], in a number of distinct tissue types. In addition to previously generated TIR1 transgenes (ubiquitously expressed *ieSi57*, pharyngeal muscle–expressed *ieSi60*, and intestine-expressed *ieSi61*) [41], we generated new transgenes that express TIR1 in tissue- or cell-specific manners (described in Materials and methods and summarized in **Fig 1D**). In the presence of auxin, these TIR1 transgenes effectively remove DAF-3/SMAD:: TagRFP-T::AID, DAF-12::TagRFP-T::AID and DAF-16::mNeonGreen::AID protein from individual cell types, as assessed by the loss of fluorescent signals (shown in **S1 Fig** and other main figures discussed later). To corroborate efficient gene depletion at a functional level, we tested whether AID-mediated protein removal, using ubiquitously expressed TIR1, can recapitulate the *daf-3/SMAD*, *daf-12/VDR*, and *daf-16/FoxO* null mutant phenotypes. To this end, we crossed the tagged loci into the *daf-7(e1372)* or *daf-2(e1370)* Daf-c mutant backgrounds that were previously shown to be suppressed by *daf-3/SMAD*, *daf-12/VDR*, or *daf-16/FoxO*

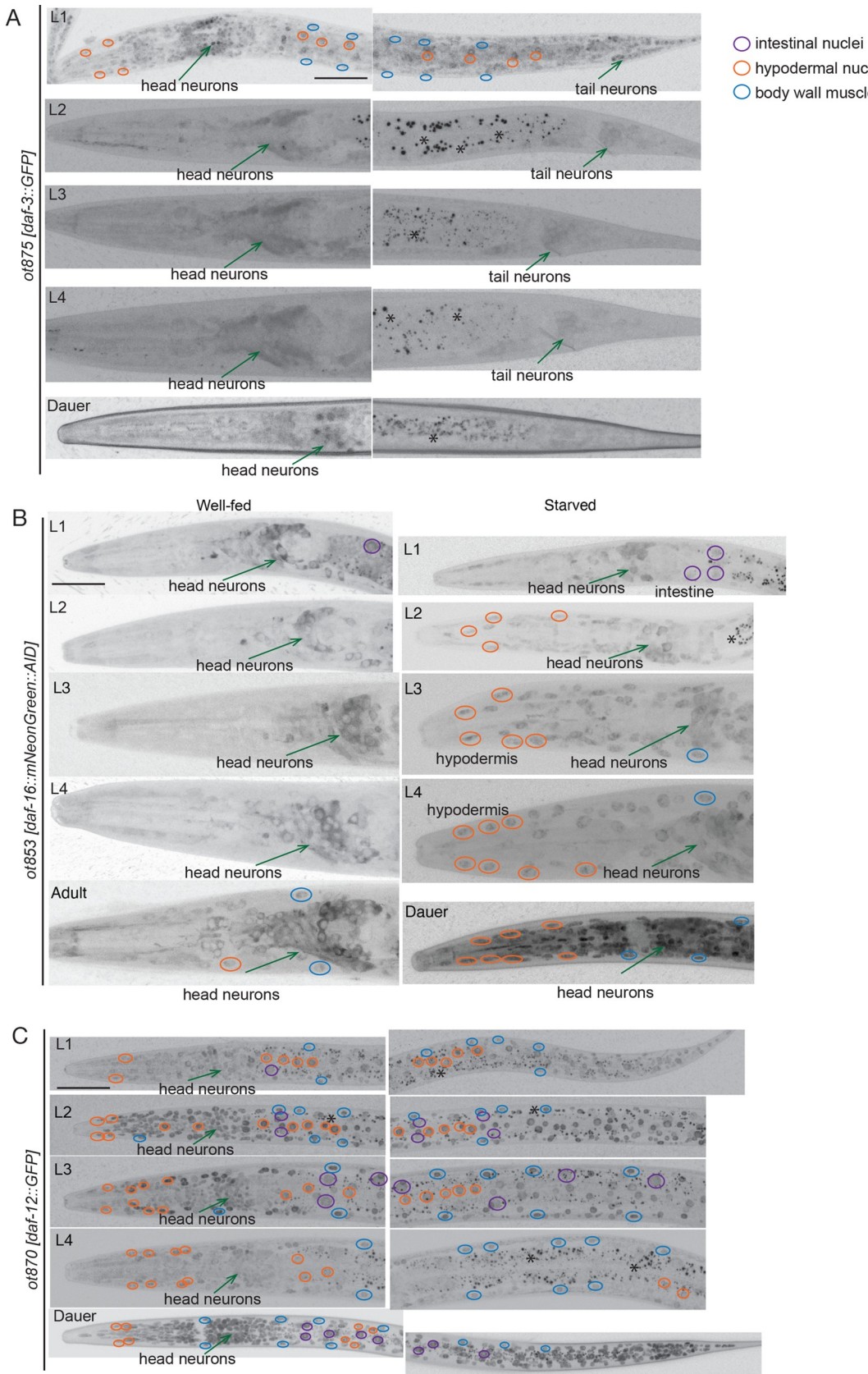

**Fig 2. Expression pattern of *daf-16/FoxO*, *daf-12/VDR*, and *daf-3/SMAD*.** (**A**) Expression of the *daf-3::GFP* CRISPR allele at different stages in development (in well-fed conditions). Anterior is to the left on all images. Scale bar, 20 μm (same for all images). (**B**) Expression of the *daf-16::mKate2* CRISPR allele at different stages in development and nutritional states. Different tissue types are indicated with color-coded circles (or an arrow, in the case of neurons). (**C**) Expression of the *daf-12::GFP* CRISPR allele at different stages in development (in well-fed conditions). AID, auxin-inducible degron.

null mutants, respectively [43–45]. Eggs of these strains were plated on control or auxin-treated plates and allowed to grow for 3 days at 25˚C (**Fig 3A**). Without ubiquitous TIR1 expression and/or without auxin addition, neither of the tagged loci suppressed the respective Daf-c mutant background, demonstrating that the reporter tagging does not affect gene function (**Figs 3B** and **4A–4C**). However, the combination of ubiquitous TIR1 expression and auxin addition resulted in a full suppression of the respective Daf-c mutant phenotypes, demonstrating the effectiveness of the AID system (**Fig 3B**).

Next, we asked whether these 3 TFs are only required for the commitment to the dauer remodeling program or whether they are continuously required to maintain the remodeled state. To address this question, we performed auxin shift experiments, in which eggs are initially plated on control plates and allowed to develop into dauers at 25˚C, which are then transferred to auxin plates on day 3 (or to auxin(-) plates for the control animals) (**Fig 3C**). Depletion of DAF-16/FoxO, DAF-3/SMAD, or DAF-12/VDR after dauer formation results in dauer exit, indicating that all 3 proteins are required for the maintenance of the remodeled dauer state (**Fig 3D**).

## Neuronal removal of DAF-3/SMAD, but not of DAF-12/VDR or DAF-16/FoxO, suppresses dauer formation

After assessing temporal requirements, we sought to define the focus of action of the 3 hormonal systems, through cell type–specific protein depletion. Previous mosaic analysis of the TGFβ receptor *daf-4* indicated that TGFβ signaling is required in the nervous system to control dauer formation [26]. In accordance with these findings, we find that depletion of DAF-3/SMAD, using a panneuronal TIR1 driver line, is indeed able to significantly suppress the DAF-3/SMAD-dependent Daf-c phenotype of *daf-7(e1372)* mutants (**Fig 4A**). To further delineate the focus of DAF-3/SMAD action, we removed DAF-3/SMAD exclusively from ciliated sensory neurons, using an *ift-20::TIR1* transgenic line, and found that this also significantly suppresses the Daf-c phenotype of *daf-7(e1372)* animals (**Fig 4A**). We conclude that DAF-3/SMAD acts in sensory neurons to nonautonomously control remodeling of multiple tissue types during dauer remodeling. This is consistent with previous transgenic rescue experiments which suggested a possible function of TGFβ receptor DAF-4 in sensory neurons in the context of controlling the expression of sensory receptor proteins [13].

Previously published transgenic overexpression approaches indicated that *daf-16/FoxO* function in either the nervous system or the gut is sufficient to rescue *daf-16/FoxO* defects [28,29], but these studies did not address in which tissue type *daf-16/FoxO* is normally required to promote dauer remodeling. Using the same panneuronal TIR1 driver line that functionally removed DAF-3/SMAD from the nervous system to suppress dauer formation, we found that panneuronal removal of DAF-16/FoxO was not able to suppress the Daf-c phenotype of *daf-2(e1370)* mutants (**Fig 4B**). Similarly, panneuronal removal of DAF-12/VDR did not suppress the Daf-c phenotype of *daf-7(e1372)* mutants (**Fig 4C**). In both cases, dauer appear morphologically like normal dauers, and they also retain their chemical resistance to SDS, a key feature of dauers (**S3 Fig**) [4]. Also, in both cases, fluorescence microscopy shows that the panneuronal TIR1 driver effectively removed DAF-16::mNG::AID and DAF-12::TagRFP-T::AID protein from the nervous system (**S1 Fig**).

An alternative focus of action for *daf-16/FoxO* has been proposed through transgenic rescue experiments, which showed that reintroduction of *daf-16/FoxO* exclusively in the intestine was

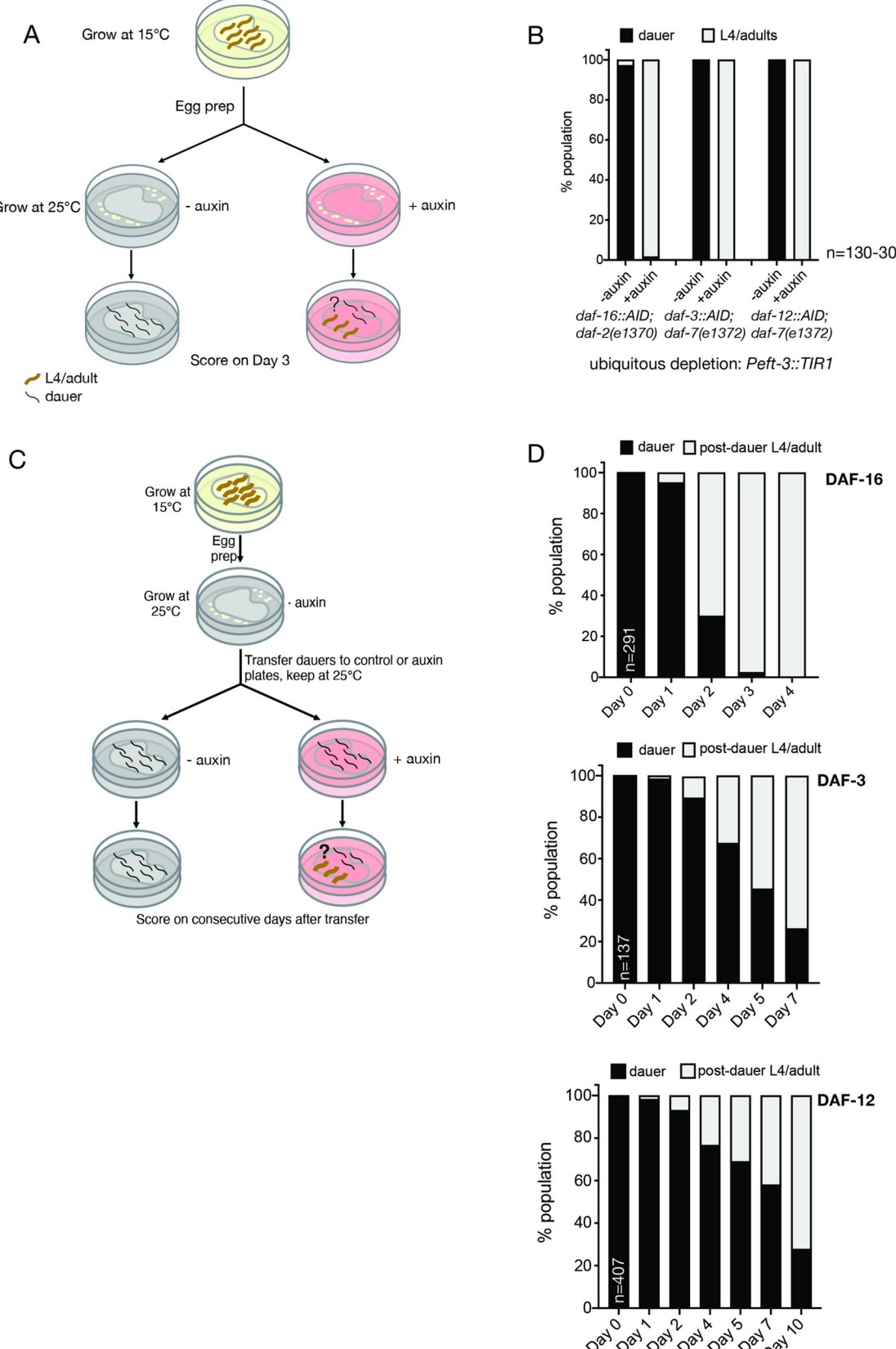

**Fig 3. Requirement of DAF-16/FoxO, DAF-12/VDR, and DAF-3/SMAD for initiation and maintenance of the dauer state.** (**A**) Schematic of the experimental design for testing conditional alleles of *daf-16*, *daf-12*, and *daf-3* for the requirement of the respective proteins in initiation of dauer formation (also applies to experiments described in later figures). (**B**) Dauer formation is suppressed upon ubiquitous depletion of DAF-16/FoxO, DAF-12/VDR, and DAF-3/SMAD, which serves as a positive control for subsequent experiments with tissue-specific protein depletion. (**C**) Schematic of the experimental design for testing conditional alleles of *daf-16*, *daf-12*, and *daf-3* for the requirement of the respective proteins in maintenance of the dauer state. (**D**) Upon ubiquitous depletion of DAF-16/FoxO, DAF-12/VDR, and DAF-3/SMAD after dauer formation, worms exit the dauer state and initiate post-dauer development, indicating the requirement of the 3 proteins in the active maintenance of the dauer state. The data underlying this figure can be found in S1 Data. AID, auxin-inducible degron.

able to revert the dauer suppression phenotype of *daf-2(e1370); daf-16(Df50)* double mutants [29]. However, our own examination of this previously published strain revealed that the suppression was not complete, namely that the transgenic dauers fail to remodel their pharynx and continue to pump, unlike *daf-2(e1370)* dauers, although they retain SDS-resistance, a distinctive characteristic of dauers (S4 Fig). To address this issue from an orthogonal angle, we examined whether intestinal depletion of DAF-16/FoxO is able to suppress the Daf-c phenotype of *daf-2(e1370)* mutants. We observed no suppression (Fig 4B), arguing that DAF-16/FoxO is at least not solely required in the intestine to promote entry into the dauer stage. We also detected no effect of DAF-12/VDR removal from the intestine on the Daf-c phenotype of *daf-7(e1372)* mutants (Fig 4C).

## Neuronal removal of DAF-16/FoxO affects dauer-specific sensory receptor expression changes

Our observation that animals that lack DAF-16/FoxO or DAF-12/VDR activity in the nervous system can still enter the dauer stage provided us with the opportunity to ask what functions DAF-16/FoxO and DAF-12/VDR play within the nervous system of dauer animals for neuronal remodeling. In the next few sections, we will first describe our analysis of DAF-16/FoxO, and at the end of this paper, we will describe our results with DAF-12/VDR.

One prominent neuronal remodeling event that relates to altered chemosensory behavior of dauers is evidenced by the many expression changes of olfactory-type G-protein–coupled receptors upon entry into the dauer stage [12–14]. For example, we previously reported that the *sri-9* GPCR gene, which is normally expressed only in the ADL neuron class in the head of the worm, becomes induced in 8 additional neuron classes upon entry into the dauer stage [14]. We find that panneuronal removal of DAF-16/FoxO largely suppresses the induction of *sri-9* reporter gene expression in these additional neuron classes (Fig 5A), suggesting that DAF-16/FoxO functions within the nervous system to control changes in GPCR expression. However, factors other than DAF-16/FoxO (or other than its function in neurons) must also play a role, since we find that the dauer-specific induction of *sra-25* in the ADL neurons [14] still occurs after panneuronal DAF-16/FoxO depletion (Fig 5B). The *daf-16*-independence of *sra-25* induction in ADL of dauers may relate to the fact that *sra-25* expression persists even after exit from the dauer stage [14], i.e., after inactivation of DAF-16 in post-dauer animals.

To extend our probing of the impact of DAF-16/FoxO on sensory receptor expression and to address the question of cell autonomy with more precise cellular resolution, we turned to another sensory receptor system, the receptor type guanylyl cyclases (rGCs). We have previously studied this gene family in the context of taste perception in non-dauer stage animals [47]. We noted that expression of one rGC, *gcy-6*, which is normally exclusively expressed in the ASEL salt receptor neuron [48], becomes activated in the AWC^OFF olfactory neuron during entry into the dauer stage (Fig 5C). We established an AWC-specific TIR1 driver line to ask whether DAF-16/FoxO acts specifically in AWC neurons to promote *gcy-6* expression during dauer remodeling and indeed found this to be the case (Fig 5C).

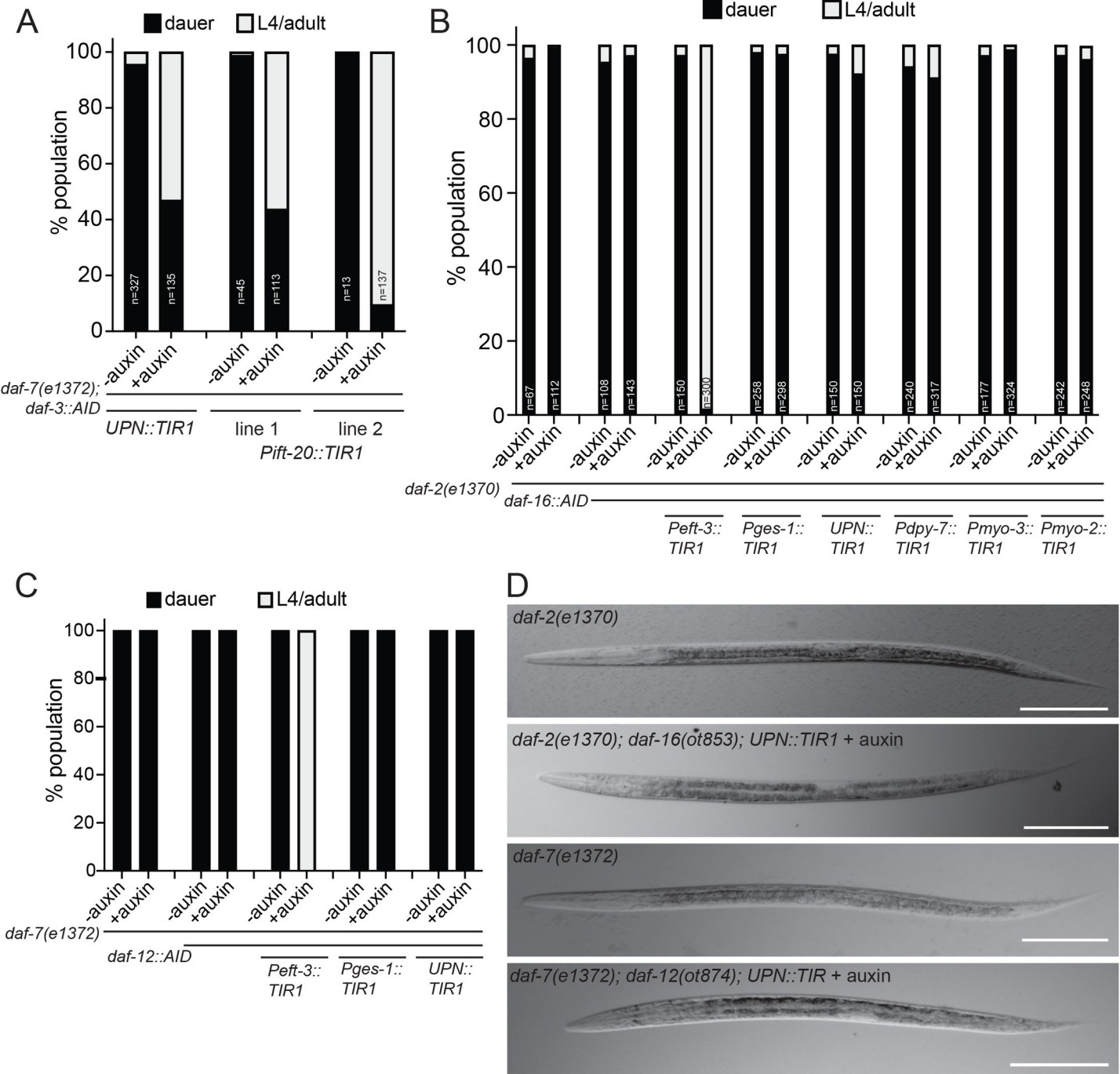

**Fig 4. Neuronal depletion of DAF-3/SMAD, but not of DAF-16/FoxO or DAF-12/VDR, suppresses dauer formation.** (**A**) Quantification of dauer formation upon depletion of DAF-3/SMAD with a panneuronal (*otIs730[UPN::TIR1::mTurquoise2]*) and a pansensory TIR1 (*otEx[Pift-20::TIR1::mRuby]*) drivers. We note that Greer et al. observed that expression of the TGFβ receptor DAF-1 in the RIM/RIC interneurons is sufficient to rescue the dauer phenotype of *daf-1* mutants [46], but these sufficiency experiments are conceptually distinct from our necessity experiments. (**B**) Quantification of dauer formation upon tissue-specific depletion of DAF-16/FoxO. (**C**) Quantification of dauer formation upon tissue-specific depletion of DAF-12/VDR. (**D**) Overall appearance of dauers with panneuronal depletion of DAF-16/FoxO and DAF-12/VDR is superficially similar to that of the background Daf-c strains, *daf-2(e1370)*, and *daf-7(e1372)*, respectively. Scale bars, 60 μm. The data underlying this figure can be found in S1 Data.

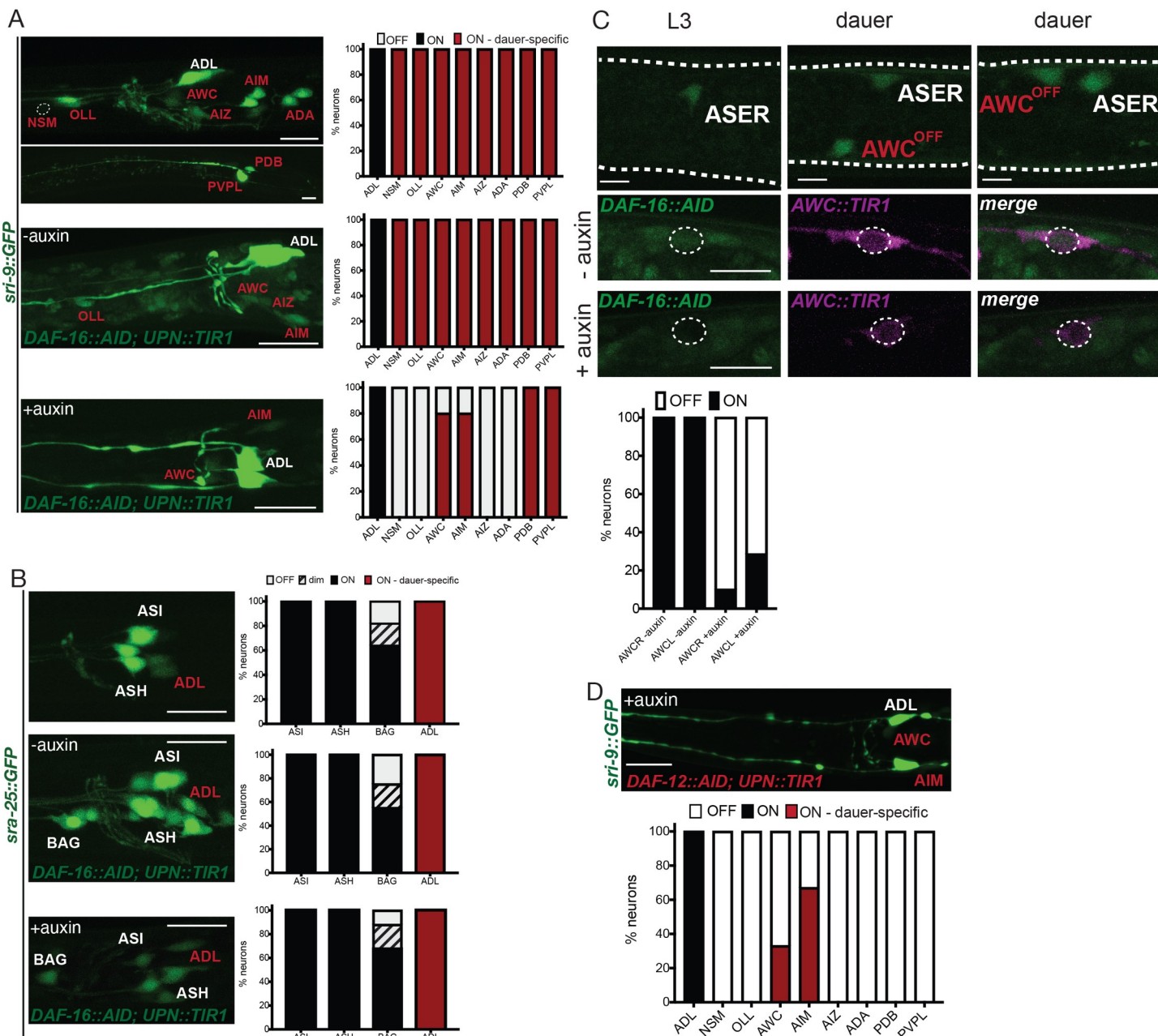

**Fig 5. Neuronal removal of DAF-16/FoxO and DAF-12/VDR affects dauer-specific sensory receptor expression changes.** (**A**) *sri-9*::*GFP* (*otIs732*) expression induced in multiple neurons in dauer (red). The dauer-specific expression is largely dependent on the neuronal function of DAF-16/FoxO. Left, head images of control and auxin-treated dauers with panneuronal depletion of DAF-16/FoxO. Right, quantification of *sri-9*::*GFP* expression in each neuronal type. *n* = 22 (control) and *n* = 10 (auxin treated). Scale bars, 10 μm. (**B**) *sra-25*::*GFP* (*sIs12199*) expression induced in ADL neurons in dauer (red). The dauer-specific expression is not dependent on neuronal function of DAF-16/FoxO. Left, head images of control and auxin-treated dauers with panneuronal depletion of DAF-16/FoxO. Right, quantification of *sra-25*::*GFP* expression. *n* = 22 (control) and *n* = 10 (auxin-treated). Scale bars, 10 μm. (**C**) *gcy-6*::*NLS*::*GFP* (*otIs586*) expression is gained in AWC[OFF] neuron in dauer. Above, images of AWC in control and auxin-treated dauers with panneuronal depletion of DAF-16/FoxO. Below, quantification of *gcy-6*::*NLS*::*GFP* expression. *n* = 12. Scale bars, 5 μm. (**D**) The dauer-specific expression of *sri-9*::*GFP* (*otIs732*) also depends on the neuronal function of DAF-12/VDR. Above, image of an auxin-treated dauer with panneuronal depletion of DAF-12/VDR. Below, quantification of *sri-9*::*GFP* expression in each neuronal type. Scale bar, 10 μm. The data underlying this figure can be found in S1 Data. AID, auxin-inducible degron.

## Cell autonomous control of electrical synapse remodeling by DAF-16/FoxO

Another prominent set of remodeling events that we recently described to occur upon dauer entry are changes in expression of electrical synapse proteins, the innexins [11]. For example, expression of the *inx-6* innexin becomes exclusively induced in the AIB interneurons of dauers. We had previously shown that this induction requires DAF-16/FoxO activity in the AIB interneurons [11]. We extended these previous findings by using the example of AIB neurons as a test case to ask whether this cell-autonomous process requires continuous DAF-16/FoxO expression. Alternatively, DAF-16/FoxO may only be transiently required, for example, as a pioneer factor to permit the induction of *inx-6* expression. By shifting the dauers initially grown in control conditions onto auxin-containing plates, we found that DAF-16/FoxO is continuously required to maintain *inx-6* expression during dauer arrest (**Fig 6A**), which is consistent with the continuous requirement for DAF-16/FoxO to retain animals in the dauer stage, as described above (**Fig 3C and 3D**).

We extended our analysis to a number of additional innexin genes, particularly also considering innexin genes whose expression does not become up-regulated, but rather becomes down-regulated upon entry into the dauer stage. One such example is *che-7*, an innexin that is up-regulated in some neurons, but is down-regulated in other neuron types upon dauer entry [11]. We find that the up-regulation of the *che-7* innexin in a number of neuron types (including BDU, NSM, and AIM) fails to occur after panneuronal depletion of DAF-16, consistent with its cell-autonomous activity (**Fig 6B and 6C, S5 Fig**). Intriguingly, the dauer-specific down-regulation of *che-7* in the PHA and OLL neurons is also abolished upon panneuronal DAF-16 depletion (**Fig 6D**), indicating that DAF-16, either directly or indirectly, activates or represses genes depending on cellular context.

However, DAF-16 is not required for the regulatory plasticity of every single innexin gene. We find that panneuronal DAF-16 is not required for the down-regulation of *inx-2*, which occurs in multiple neurons types upon dauer entry (**Fig 6E**).

## Remodeling of locomotory behavior requires neuronal DAF-16/FoxO

Entry into the dauer stage is accompanied by remarkable changes in locomotory behavior [4,11,17]. Among such changes are a decrease in the wave amplitude of the sinusoidal movement patterns of the animals and a decrease in various aspects of reversal behavior, including omega turn behavior, as quantified using a semiautomated WormTracker system [11]. We used the same system to quantify locomotor aspects of dauer-arrested animals in which DAF-16/FoxO was panneuronally depleted. Strikingly, these animals retain many patterns of non-dauer locomotion, i.e., they fail to undergo the alterations normally associated with dauer behavior. For example, among other locomotory features, panneuronally DAF-16/FoxO-depleted dauer animals display an increased amplitude of sinusoidal movement and an increase in omega turn frequency (**Fig 7A and 7B**). These animals move much faster than vehicle controls as well as *daf-2(e1370)* dauers and pause less during locomotion. Comparison with fed L3 larvae of the *daf-2(e1370)* background strain show that panneuronally DAF-16-depleted dauers behave more L3-like than bona fide dauers (**Fig 7A and 7B**). Depleting DAF-16 from body wall muscle did not result in any significant changes in locomotory behavior (**S6 Fig**), but, as we will describe further below, it did have other effects on the animal.

## Remodeling of the pharynx reveals cell autonomous and nonautonomous function of DAF-16/FoxO

Another conspicuous behavioral change observed upon entry into the dauer stage is the cessation of pharyngeal pumping [4]. Pharyngeal pumping is controlled by the pharyngeal nervous

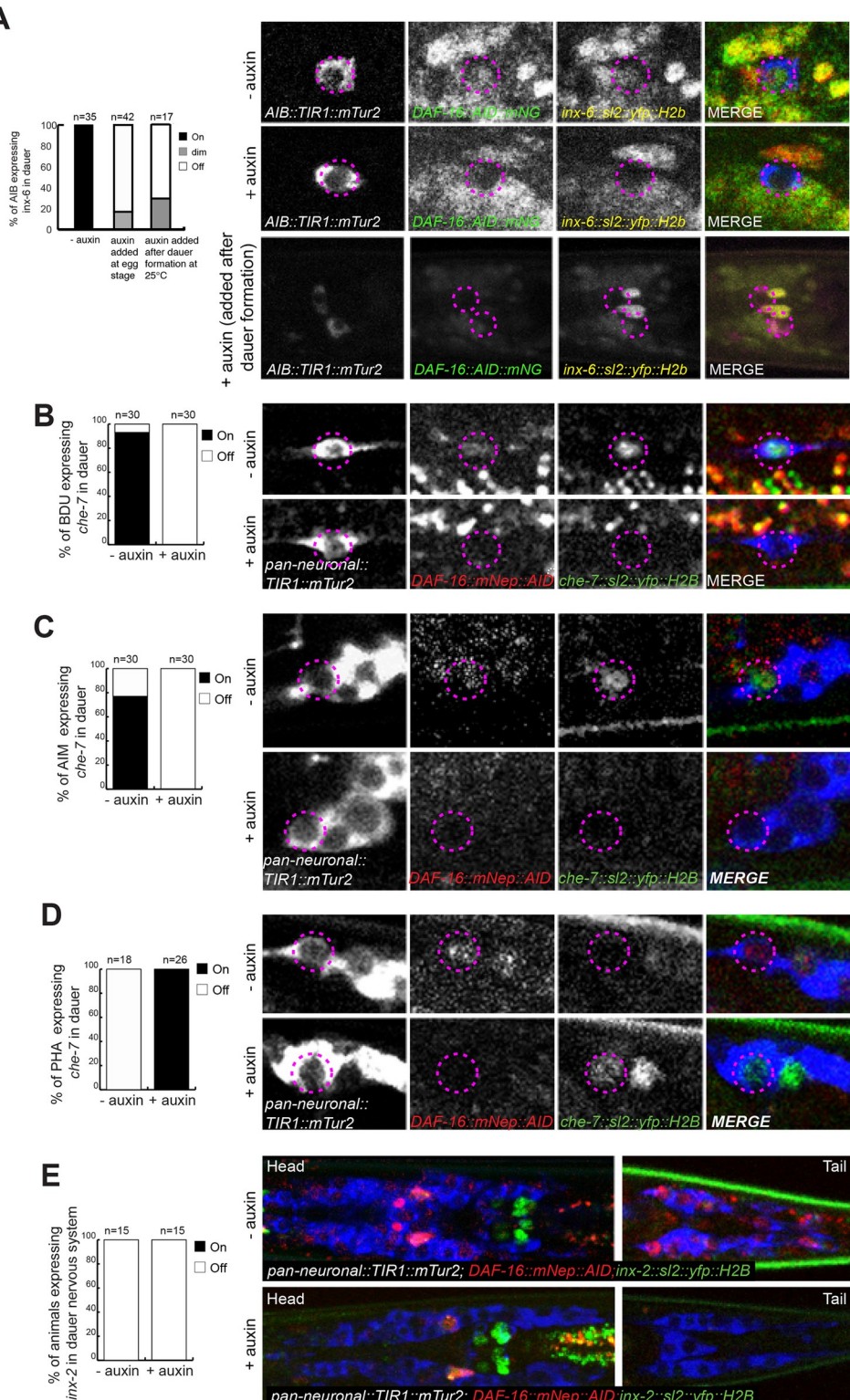

**Fig 6. Neuronal removal of DAF-16/FoxO controls dauer-specific expression changes of electrical synapse components.** (**A**) The dauer-specific expression gain of an *inx-6* reporter allele *(ot804)* in AIB is affected upon continuous AIB-specific depletion of DAF-16/FoxO, as well as in animals where DAF-16/FoxO is depleted in AIB after dauer remodeling. (**B**) A *che-7* reporter (*otEx7112*) expression is gained in BDU neurons in dauer. This dauer-specific *che-7* expression is lost upon panneuronal depletion of DAF-16/FoxO in auxin-treated dauers. (**C**) A *che-7* reporter

(*otEx7112*) expression is gained in AIM neurons in dauer. This dauer-specific *che-7* expression is lost upon panneuronal depletion of DAF-16/FoxO in auxin-treated dauers. **(D)** A *che-7* reporter (*otEx7112*) expression is down-regulated in PHA neurons in dauer. This dauer-specific down-regulation of *che-7* expression is affected upon panneuronal depletion of DAF-16/FoxO in auxin-treated dauers. **(E)** Expression of an *inx-2* reporter allele (*ot906*) is down-regulated in multiple neurons in dauer. This dauer-specific down-regulation of *inx-6* expression is unaffected upon panneuronal depletion of DAF-16/FoxO in auxin-treated dauers. In all the animals scored, AID-tagged DAF-16/FoxO was also completely depleted in the presence of auxin. The data underlying this figure can be found in S1 Data. AID, auxin-inducible degron.

system, composed of 14 neuron classes that are heavily interconnected and that innervate pharyngeal musculature [49–51]. The cessation of pharyngeal pumping during dauer remodeling is also accompanied by an autophagy-dependent shrinkage of pharyngeal muscle tissue [52], thereby resulting in a constricted appearance of the dauer pharynx. We find that panneuronal depletion of DAF-16/FoxO results in a derepression of pharyngeal pumping in dauer animals (**Fig 8A**), indicating that DAF-16/FoxO is normally required in neurons to suppress pumping. Unexpectedly, neuronal depletion of DAF-16/FoxO also results in a failure to constrict pharyngeal musculature (**Fig 8B and 8C**). To further explore this observation, we also generated a pharyngeal muscle–specific TIR1 driver line and found that pharyngeal muscle depletion of DAF-16/FoxO likewise resulted in a loss of pharyngeal muscle constriction as well as derepression of pharyngeal pumping upon dauer remodeling. Muscle constriction and pharyngeal pumping are not obligatorily linked because in hypomorphic *daf-16(m26)* mutants, pharyngeal muscle tissue of dauer animals fails to constrict [53], but pharyngeal pumping remains suppressed (**S7A and S7B Fig**). Vice versa, wild-type dauers recovering on food show pharyngeal pumping within 2 to 3 hours after food introduction (**S7C Fig**), even though their pharynx is still constricted.

To further dissect the focus of neuronal action of *daf-16*, we depleted DAF-16 exclusively from pharyngeal neurons, using a *cis*-regulatory element from the *ehs-1* gene [39] to drive TIR1 expression in pharyngeal, but no other neurons. We found that TIR1 expression driven by this *ehs-1* promoter fragment was strong enough in all the pharyngeal neurons, except I1 and I4, to deplete DAF-16 in the presence of auxin (**Fig 9A and 9G**). This also resulted in derepression of pharyngeal pumping in dauers (**Fig 9B**). The *ehs-1* promoter fragment used here is also expressed in the head muscle of L1 larvae, but we find that depletion of DAF-16 from all body wall muscles, using the *myo-3* promoter, does not affect pharyngeal activity of dauers (**Fig 8A**). Hence, the effect on pharyngeal pumping seen with the *ehs-1* promoter fragment can be fully attributed to pharyngeal neurons themselves.

To further pinpoint the focus of action of DAF-16 in the pharyngeal nervous system, we removed DAF-16 from specific subpopulations of pharyngeal neurons, using driver lines that express TIR1 in either the I1, M1, M2, M4, M5, and MC cholinergic neurons of the pharynx (*unc-17prom4* driver) or in I5 and M3 glutamatergic neurons of the pharynx (*eat-4prom7* driver) (**Fig 9G**) [38]. We confirmed DAF-16 removal from these pharyngeal neurons by imaging the loss of the fluorescent protein signal of DAF-16::mNG::AID (**Fig 9C and 9E**). We find that removal of DAF-16 from either neuronal population partially derepressed pumping (**Fig 9D and 9F**). Since DAF-16 was depleted from nonoverlapping neurons in these 2 TIR1 driver lines (**Fig 9G**), this suggests that DAF-16 function is distributed over different pharyngeal neuron types to silence pumping of the pharynx in the dauer stage.

We conclude that DAF-16/FoxO function is required in 2 distinct tissues to control behavioral and structural remodeling of the pharynx, the nervous system as well as pharyngeal muscle. As detailed in the next section, we found yet another tissue type that requires DAF-16/FoxO for pharynx remodeling.

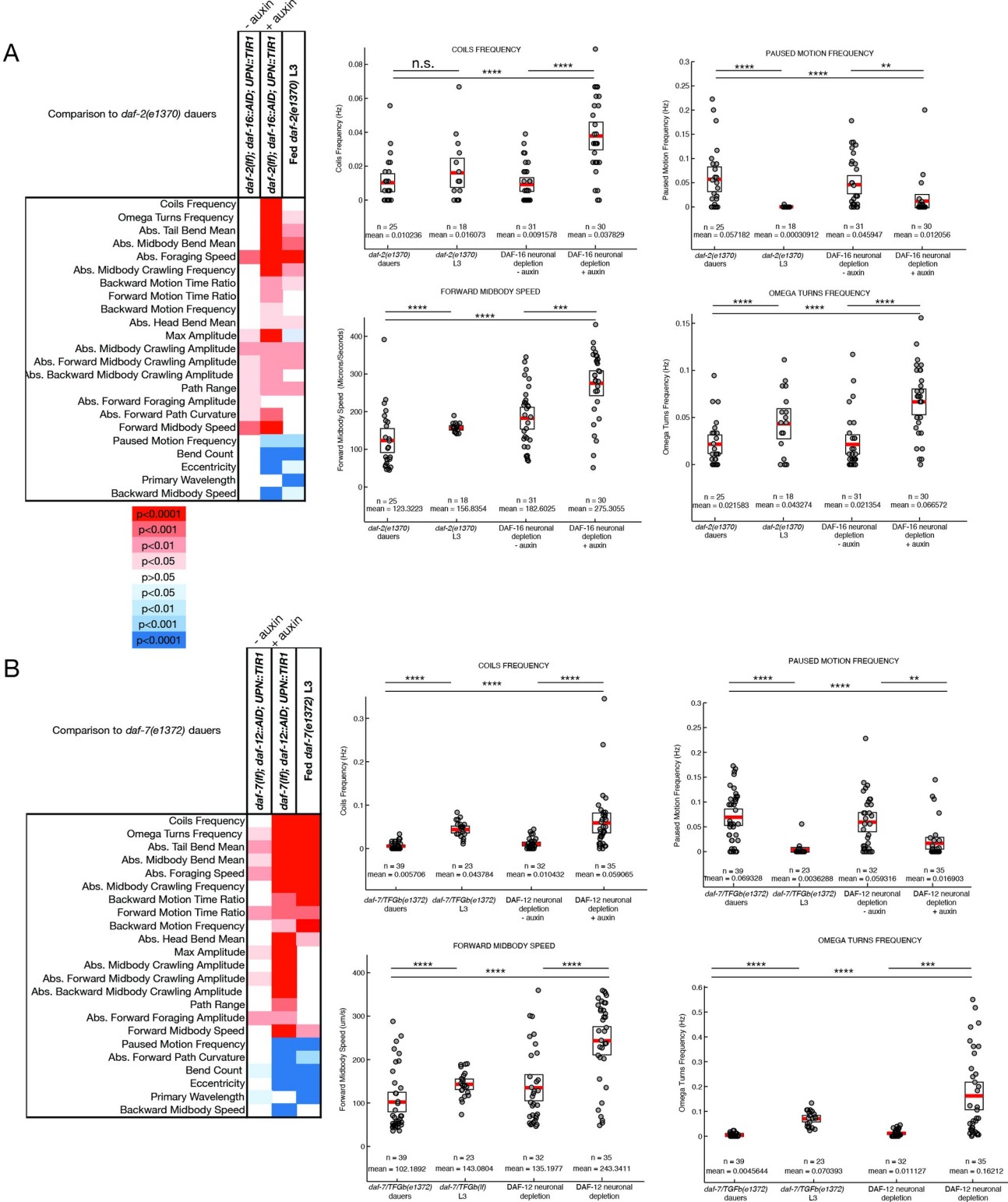

**Fig 7. DAF-16/FoxO and DAF-12/VDR act in the nervous system to affect dauer-specific locomotory behavior.** (**A**) A heat map of *p*-values associated with locomotory features of panneuronally DAF-16/FoxO-depleted dauers. (**B**) Representative locomotory features, which distinguish dauers with panneuronal depletion of DAF-16/FoxO from controls. (**C**) A heat map of *p*-values associated with locomotory features of panneuronally DAF-12/VDR-depleted dauers. (**D**) Representative locomotory features which distinguish dauers with panneuronal depletion of DAF-12/VDR from controls. Each circle

represents the experimental mean of a single animal. Red lines indicate the mean of means, and rectangles indicate SEM. Wilcoxon rank-sum tests and false discovery rate q values for each comparison: n.s., nonsignificant, $*q < 0.05$, $**q < 0.01$, $***q < 0.001$, $****q < 0.0001$. The data underlying this figure can be found in S2 Data.

## Intestinal depletion of DAF-16/FoxO has cell-autonomous and cell nonautonomous consequences

We further explored autonomous and nonautonomous functions of DAF-16/FoxO by depleting it from the intestine using a *ges-1*-driven TIR1 transgene. In a *daf-2* mutant background, these transgenic animals still enter the dauer stage, indicating that DAF-16/FoxO is not required in the intestine for dauer entry (**Fig 4B**). The lack of an effect on dauer formation allowed us to study the function of DAF-16/FoxO in the intestine of dauer stage animals.

Using Oil Red O staining, we observed that DAF-16/FoxO is required in the intestine to control the metabolic remodeling of intestinal cells observed upon entry into the dauer stage (**Fig 10A and 10B**). This is consistent with previous studies that utilized transgenic rescue approaches [54] and with the observation that the promoter of intestinally expressed fat metabolism genes are directly targeted by DAF-16/FoxO [55,56].

Loss of function mutants in the insulin/IGF receptor ortholog DAF-2 exhibit longer life span than wild-type animals, an effect that depends on DAF-16/FoxO [43]. We recapitulated this effect through DAF-16::AID removal with a ubiquitously expressed TIR1 driver (**Fig 10C**). Complementing previous transgenic rescue studies [28], we find that depletion of DAF-16/FoxO exclusively from the intestine strongly suppresses the life span extension of *daf-2 (e1370)* mutants (**Fig 10C**), while muscle-specific *daf-16* removal does not (**Fig 10C**). We noted that DAF-16::mNG::AID animals already have a reduced life span on their own even in the absence of TIR1 but nevertheless, in the presence of ubiquitous or intestinal TIR1 and auxin its effects on *daf-2* longevity become strongly enhanced (**Fig 10C**). A tight dosage sensitivity of *daf-16* in the determination of life span has been reported before [57].

Unexpectedly, we discovered that in addition to its metabolic and life span effect, intestinal DAF-16/FoxO depletion also results in a failure to remodel pharyngeal pumping behavior and pharyngeal muscle constriction (**Fig 8A and 8B**). By auxin-shifting the dauers initially grown in control conditions, we find that the DAF-16/FoxO requirement in the gut is continuous, since removal of DAF-16/FoxO in the intestine post-dauer entry still results in derepression of pharyngeal pumping (**Fig 9H and 9I**). Taken together, DAF-16/FoxO is required in 3 different tissue types to control pharyngeal remodeling, one of them (the intestine) being completely external to the pharynx itself.

We also asked whether intestinal DAF-16/FoxO has other nonautonomous defects in the nervous system. We specifically asked whether the down-regulation of *inx-2*, which we found not to require neuronal DAF-16/FoxO (see above; **Fig 6E**), may require intestinal DAF-16/FoxO. We found this not to be the case (**S5 Fig**). Similarly, intestinal DAF-16/FoxO depletion does not affect the neuronal changes of *che-7* and *inx-6* induction in neuronal cells (**S5 Fig**).

## Neuronal DAF-16/FoxO depletion affects the intestine

The experiments described so far provide evidence for the gut providing signals to other tissues. We also found that the intestine is at the receiving end of DAF-16/FoxO-mediated non-cell autonomous function in other tissues. Specifically, panneuronal depletion of DAF-16/FoxO affects the size of Oil Red O-positive droplets in the intestine (**Fig 10B**). In these animals, intestinal DAF-16/FoxO still translocated normally to the nucleus (**S1 Fig**), indicating that (a) neuronal DAF-16/FoxO is not providing insulin-mediated signals to promote nuclear

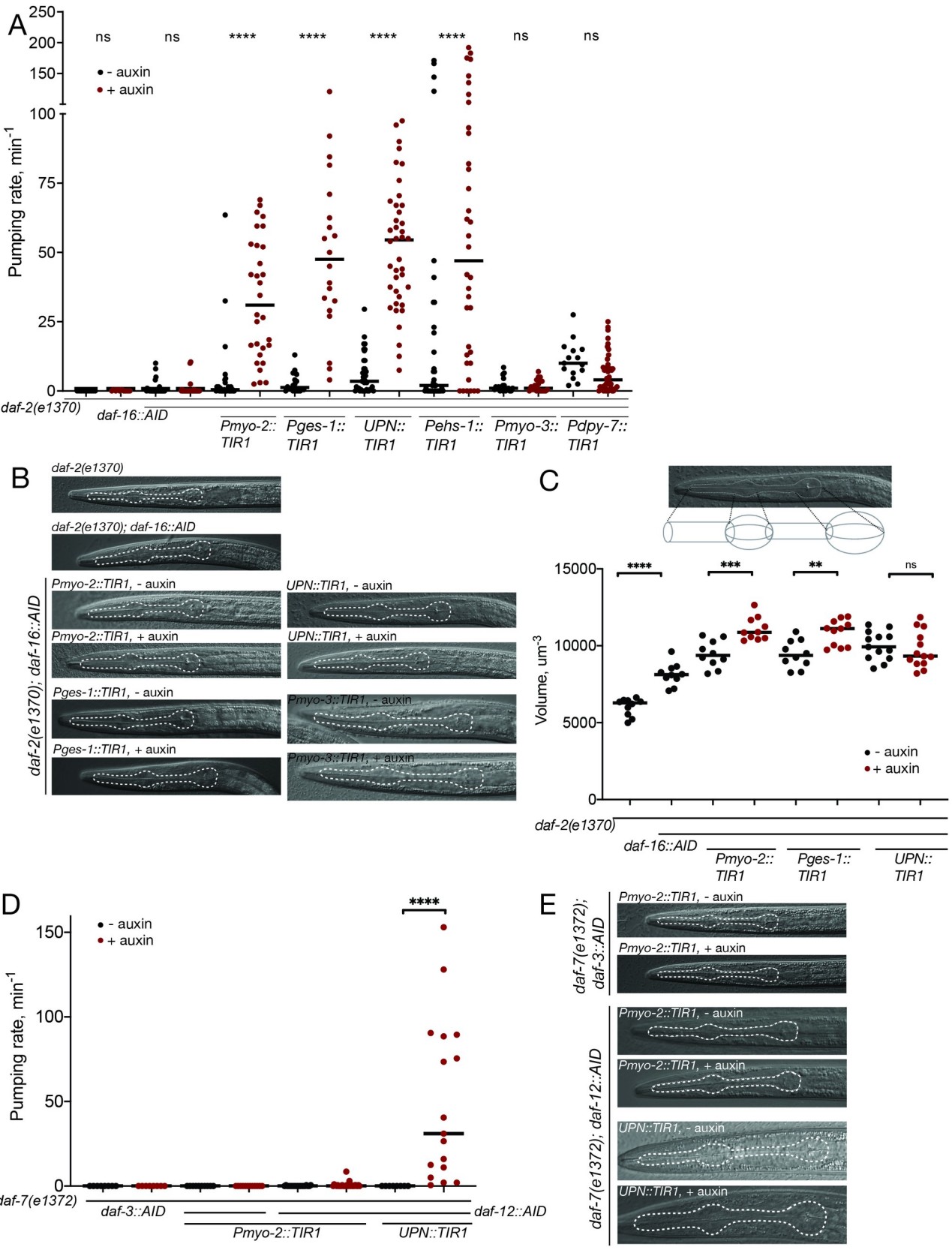

**Fig 8. Control of pharyngeal pumping by DAF-16/FoxO and DAF-12/VDR.** (**A**) Pharyngeal silencing in dauer depends on cell-autonomous and nonautonomous DAF-16 function. Quantification of pharyngeal pumping upon DAF-16/FoxO depletion in pharyngeal muscle (*myo-2*), intestine (*ges-1*), all neurons (*UPN*), pharyngeal neurons (*ehs-1*), body wall muscle (*myo-3*), and hypodermis (*dpy-7*). (**B**) Pharyngeal constriction depends on cell-autonomous and nonautonomous DAF-16/FoxO function. DIC images of control dauers and dauers with DAF-16/FoxO depleted from pharyngeal muscle, intestine, neurons, and body wall muscle. (**C**) Quantification of changes in pharynx volume upon depletion of DAF-16 from pharyngeal muscle, intestine, and neurons. (**D**) Pharyngeal silencing in dauer depends on the neuronal function of DAF-12. Quantification of pharyngeal pumping upon depletion of DAF-12/VDR in pharyngeal muscle and neurons, as well as upon depletion of DAF-3/Smad in pharyngeal muscle. (**E**) Pharyngeal constriction in dauer depends on neuronal function of DAF-12/VDR. DIC images of control dauers and dauers with DAF-12/VDR depleted from pharyngeal muscle and neurons as well as with DAF-3/Smad depleted from pharyngeal muscle. The data underlying this figure can be found in S1 Data. AID, auxin-inducible degron; DIC, differential interference contrast.

translocation of intestinal DAF-16/FoxO; and (b) that intestinal nuclear translocation of DAF-16/FoxO is alone not sufficient to affect Oil Red O droplets size in the intestine.

## DAF-16/FoxO acts cell-autonomously for muscle remodeling

Muscles also undergo remodeling upon entry in the dauer stage. They extend additional muscle arms into the nerve cords to receive supernumerary synaptic inputs from motor neurons, leading to an increased sensitivity to neurotransmitter signaling [58,59]. Using a muscle-specific TIR1 driver line, we find that DAF-16/FoxO is required in muscle during dauer remodeling to generate these supernumerary muscle arms (**Fig 11A and 11B**).

Body wall muscle of dauer animals dramatically shrink, as determined by electron microscopy (https://www.wormatlas.org/dauer/muscle/Musframeset.html). We find that muscle-specific depletion of DAF-16/FoxO affected overall dauer morphology, resulting in a widening of the body and a slight increase in body length (**S6A–S6C Fig**). This observation is consistent with DAF-16/FoxO acting cell-autonomously in muscle to affect muscle shrinkage and hence, overall body width. As mentioned above, muscle-specific removal of DAF-16/FoxO has no effect on locomotion of the animals (**S6 Fig**).

## Insulin-FoxO signaling is sufficient to suppress pharynx pumping

The above DAF-16/FoxO depletion experiments demonstrate that DAF-16/FoxO is required in a number of distinct tissue types to suppress tissue remodeling. We asked whether insulin signaling–controlled DAF-16/FoxO may also be sufficient to control remodeling. To this end, we generated a dominant negative version of the insulin/IGF-like receptor DAF-2 ("DAF-2$^{DN}$") by replacing its intracellular kinase domain with a fluorophore (**Fig 12A**). Due to the dimerization properties of insulin/IGF-like receptors [61], DAF-2$^{DN}$ is predicted to antagonize endogenous receptor signaling through the formation of an inactive dimer; this should then result in nuclear translocation of DAF-16/FoxO. We overexpressed this DAF-2$^{DN}$ construct in 3 different tissue types, in which we had found DAF-16/FoxO to be required to down-regulate pharyngeal pumping (panneuronal, intestinal, and pharyngeal muscle). We find that in young adult animals, in which we express DAF-2$^{DN}$ panneuronally, DAF-16/FoxO partially translocates to neuronal nuclei, as expected (**Fig 12B**). These animals display a significantly reduced pharyngeal pumping rate (**Fig 12C**), demonstrating that the inhibition of insulin signaling and hence activation of DAF-16/FoxO is sufficient to down-regulate pumping, even outside of dauer context. The same phenotypic consequences are observed upon intestinal expression of DAF-2$^{DN}$. In these animals, DAF-16/FoxO translocates to the nucleus in the intestine, and there is a concomitant down-regulation of pharyngeal pumping (**Fig 12B and 12C**). Notably, pharyngeal muscle expression of DAF-2$^{DN}$ does not trigger nuclear translocation in pharyngeal muscle (**Fig 12B**). Consistent with this, pharyngeal muscle-expressed DAF-2$^{DN}$ does not result in a reduction of pharyngeal pumping (**Fig 12C**).

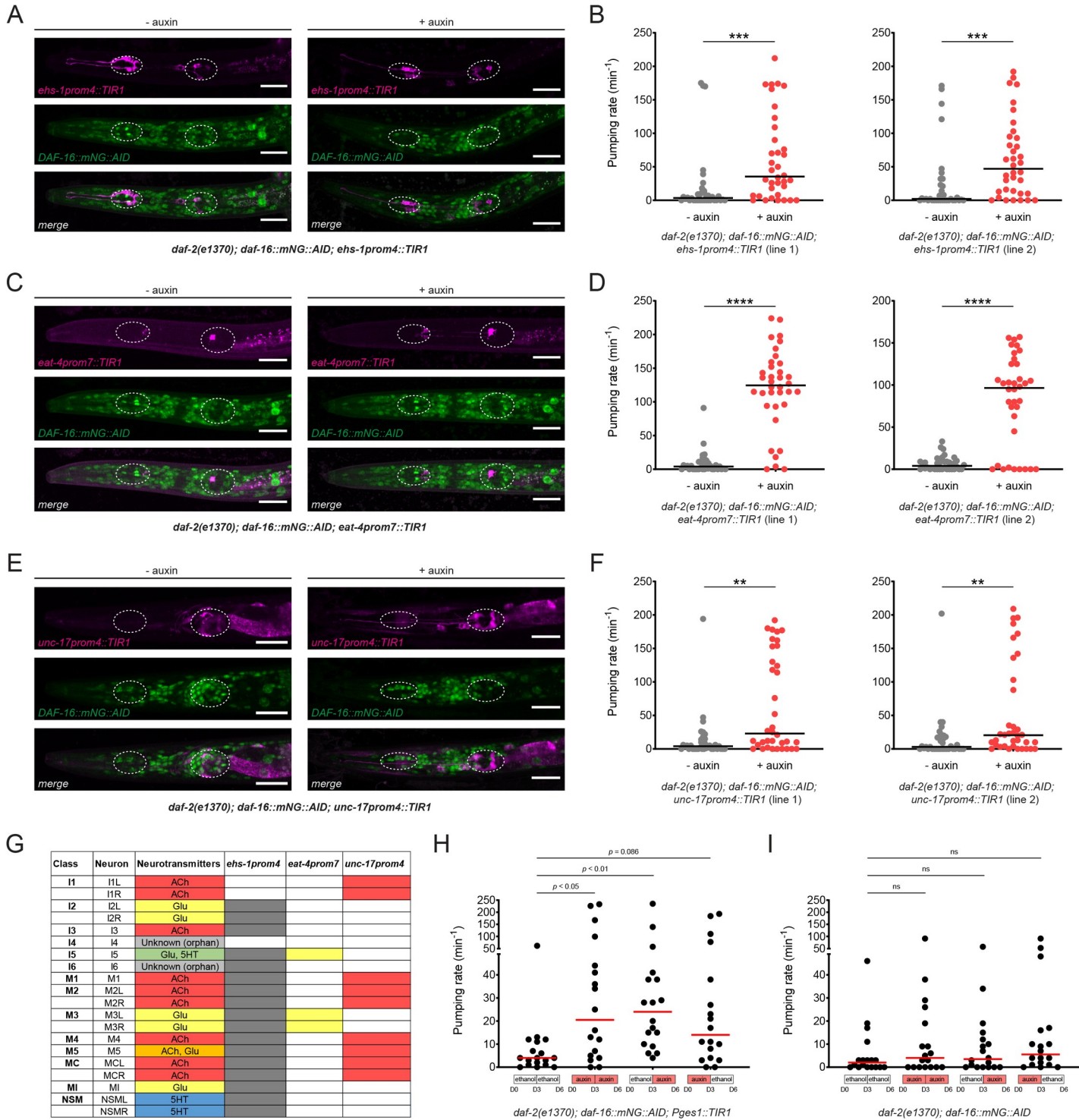

**Fig 9. DAF-16/FoxO activity is distributed across multiple pharyngeal neurons to silence pharyngeal pumping in the dauer stage.** (**A**) DAF-16/FoxO depletion in pharyngeal neurons of *daf-2(e1370)* dauers using *ehs-1prom4*-driven TIR1. In 20/20 animals observed, AID-tagged DAF-16 was not detectable in TIR1-expressing neurons (all pharyngeal neurons except I1 and I4) after auxin treatment. Scale bars, 20 μm. (**B**) Pharyngeal pumping rate after DAF-16/FoxO depletion in *daf-2(e1370)* dauers using *ehs-1prom4*-driven TIR1 (*otEx7628* and *otEx7629*). Horizontal lines represent median for 36 animals per condition. *** indicates $p < 0.001$ in Mann–Whitney test. (**C**) DAF-16/FoxO depletion in glutamatergic pharyngeal neurons of *daf-2(e1370)* dauers using *eat-4prom7*-driven TIR1. In 20/20 animals observed, AID-tagged DAF-16 was not detectable in TIR1-expressing pharyngeal neurons (I5 and M3) after auxin treatment. Scale bars, 20 μm. (**D**) Pharyngeal pumping rate after DAF-16/FoxO depletion in *daf-2(e1370)* dauers using *eat-4prom7*-driven TIR1 (*otEx7670* and *otEx7671*). Horizontal lines represent median for 36 animals per condition. **** indicates $p < 0.0001$ in Mann–Whitney test. (**E**) DAF-16/FoxO depletion in cholinergic pharyngeal neurons of *daf-2(e1370)* dauers using *unc-17prom4-*

driven TIR1. In 20/20 animals, AID-tagged DAF-16 was not detectable in TIR1-expressing pharyngeal neurons (I1, M1, M2, M4, M5, and MC) after auxin treatment. Scale bars, 20 μm. (**F**) Pharyngeal pumping rate after DAF-16/FoxO depletion in *daf-2(e1370)* dauers using *unc-17prom4*-driven TIR1 (*otEx7668* and *otEx7669*). Horizontal lines represent median for 36 animals per condition. ** indicates $p < 0.01$ in Mann–Whitney test. (**G**) Neurotransmitter identity of all 14 pharyngeal neurons. Shaded boxes for each promoter fragment indicate the neurons in which TIR1 expression is strong enough to completely deplete AID-tagged DAF-16/FoxO in the presence of auxin (20 animals analyzed per strain). (**H and I**) Pharyngeal pumping rate in *daf-2(e1370)* dauers transferred from ethanol (solvent) to auxin or from auxin to ethanol after dauer entry (dauers were transferred on day 3 after hatching, and pumping rate was measured on day 6 after hatching). DAF-16/FoxO was depleted in the intestine (*Pges1::TIR1*) or in no tissue (no TIR1 control) in the presence of auxin. Horizontal red lines represent median for 18 animals per condition. *p*-Values are for Dunn's multiple comparisons test performed after 1-way ANOVA on ranks. The data underlying this figure can be found in S1 Data. AID, auxin-inducible degron.

As a complementary approach to the DAF-2$^{DN}$ experiments, we also generated transgenic animals that express an activated, insulin signaling–independent form of DAF-16/FoxO exclusively in the intestine. This activation was achieved by mutating 4 residues (serine or threonine) that are normally phosphorylated by Akt kinases to prevent DAF-16/FoxO nuclear translocation in fed animals [27,62]. We confirmed that in these animals, mutated DAF-16/FoxO is indeed now localized to the nucleus (**Fig 12D**), and we find that in normal, non-starved adult animals, pharyngeal pumping is significantly reduced (**Fig 12E**). We conclude that DAF-2/DAF-16 signaling is not only required but also sufficient to control cellular remodeling events.

## The nuclear hormone receptor DAF-12/VDR also controls neuronal remodeling

Even though the nuclear hormone receptor *daf-12/VDR* has previously been shown to be essential for dauer remodeling [24,63], there have been no studies that address its focus of action for the dauer decision and/or dauer-specific tissue remodeling events. We first asked whether and where DAF-12/VDR is required for the remodeling of locomotory behavior in dauer animals. We used the same strategy as for DAF-16/FoxO and found that panneuronal depletion of DAF-12/VDR results in striking deviations from control dauer locomotion, even stronger than those observed upon panneuronal depletion of DAF-16/FoxO (**Fig 7C and 7D**). Specifically, panneuronally DAF-12/VDR-depleted dauers crawl faster, pause less, coil and turn more frequently, and have a flatter waveform (decreased wave amplitude) than control dauers (**Fig 7C and 7D**, **S6A and S6B Fig**). All these features make them more L3-like, as a comparison with L3 larvae of the background *daf-7(e1372)* strain shows. The magnitude of effect in many cases is larger than that of DAF-16/FoxO panneuronal depletion (**Fig 7B and 7D**).

Panneuronal DAF-12/VDR depletion also affects pharyngeal pumping and alterations in GPCR expression profiles. As with the locomotory effect, DAF-12/VDR removal shows some, but not complete overlaps in cellular specificity with DAF-16/FoxO removal. Dauer-specific expression changes of *sri-9* require DAF-16/FoxO and DAF-12/VDR in the NSM, OLL, AIZ, and AWC neurons (**Fig 5A and 5D**). However, DAF-12/VDR is not required for *sri-9* expression changes in the ADA and AIM neurons (**Fig 5D**).

There are also cell nonautonomous functions of neuronal DAF-12/VDR. We find that DAF-12/VDR removal from the nervous system affects pharyngeal muscle constriction and intestinal metabolic remodeling (**Figs 8D, 8E, and 10D**). However, unlike the case of DAF-16/FoxO, pharyngeal muscle depletion of DAF-12/VDR does not affect pharyngeal pumping or pharyngeal muscle constriction (**Fig 8D and 8E**).

As in the case with DAF-16/FoxO, intestinal DAF-12/VDR depletion does not prevent dauer formation (**Fig 4C**). However, in contrast to the intestinal DAF-16/FoxO depletion, intestinal DAF-12/VDR depletion does not affect pharyngeal muscle constriction or

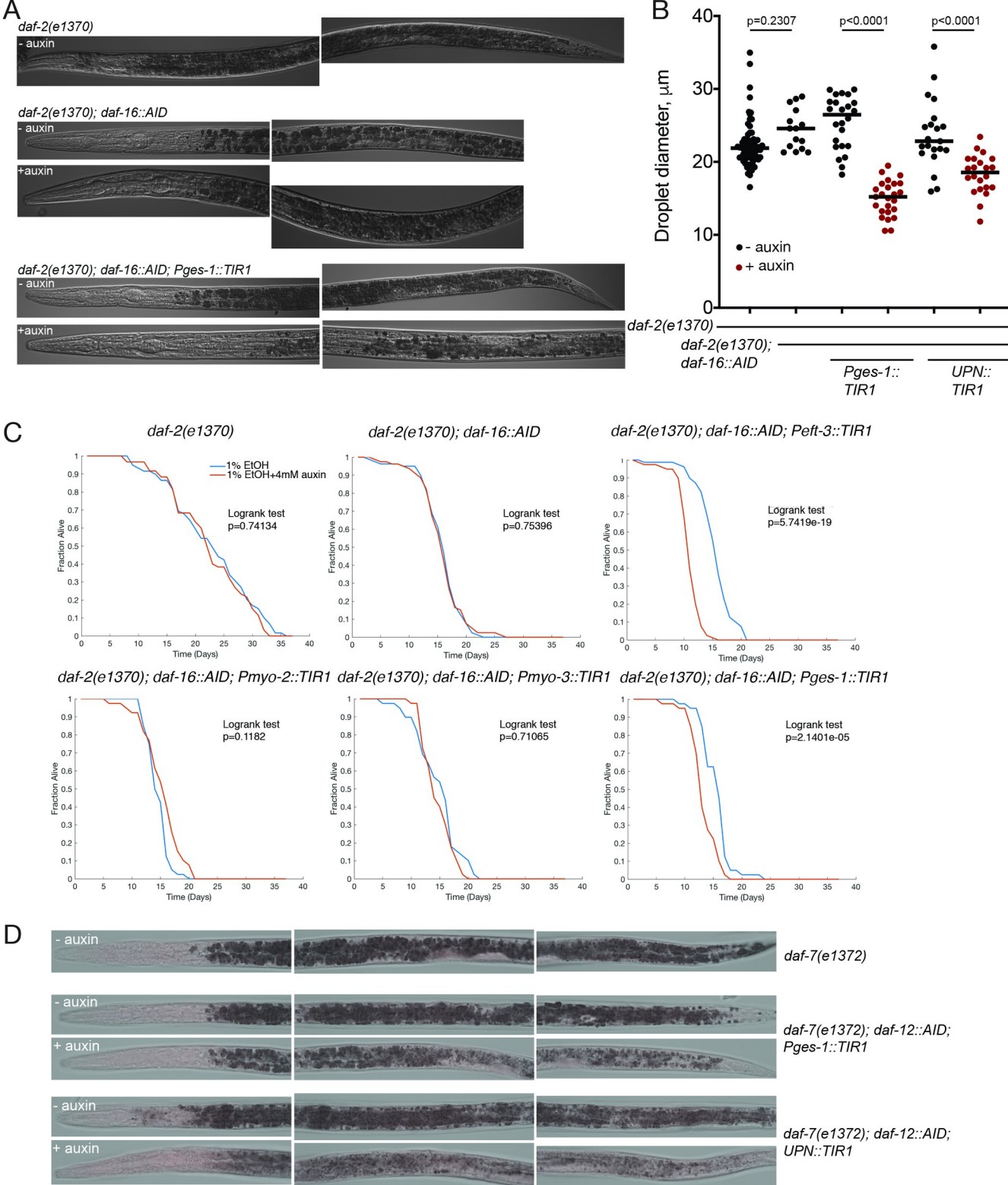

**Fig 10. Intestinal removal of DAF-16/FoxO and DAF-12 has cell autonomous and cell nonautonomous consequences.** (**A**) Oil Red O staining of control and intestinal DAF-16/FoxO-depleted strains. Note a decrease in the intensity of staining upon auxin treatment of the strain with intestinal depletion of DAF-16/FoxO. (**B**) Quantification of lipid droplet size for strains with intestinal and panneuronal depletion of DAF-16/FoxO reveals autonomous and nonautonomous regulation of dauer lipid metabolism. (**C**) Survival curves for control and auxin treatment conditions for the strains of the indicated

genotypes. (**C1-2**) Long life span of *daf-2(e1370)* mutants requires DAF-16. (**C3**) Ubiquitous depletion of DAF-16/FoxO strongly reduces life span in a *daf-2 (e1370)* background. (**C4-5**) Depletion of DAF-16/FoxO in pharyngeal or body wall muscles did not significantly reduce life span in a *daf-2(e1370)* background. (**C6**): Intestinal depletion of DAF-16/FoxO strongly reduces life span in a *daf-2(e1370)* background. $N = 40–80$ worms for each group. (**D**) Oil Red O staining of control and intestinal and panneuronal DAF-12/VDR-depleted strains. Panneuronal DAF-12/VDR depletion has a stronger effect on dauer lipid reserves. The data underlying this figure can be found in S1 Data. AID, auxin-inducible degron.

pharyngeal pumping. But intestinal DAF-12/VDR depletion does affect Oil red O straining in the intestine, albeit not as strongly as panneuronal DAF-12/VDR depletion does (**Fig 10B**).

Taken together, similar to DAF-16/FoxO, DAF-12/VDR displays cell-autonomous functions in a number of neuronal remodeling events, but also has nonautonomous roles, with the phenotypic spectrum of the tissue-specific TF depletion overlapping in some, but not in other, cases.

## The cellular specificity of the effect of the broadly expressed hormonal systems is controlled by terminal selector-type transcription factors

DAF-16/FoxO and DAF-12/VDR are both broadly expressed, but the readouts of their activity are highly cell type specific. For example, GPCR- or innexin-encoding genes are turned on in a DAF-16-dependent manner in only specific sets of neurons. To address the cell type and target specificity of the effects of DAF-16/FoxO and DAF-12/VDR, we considered a potential

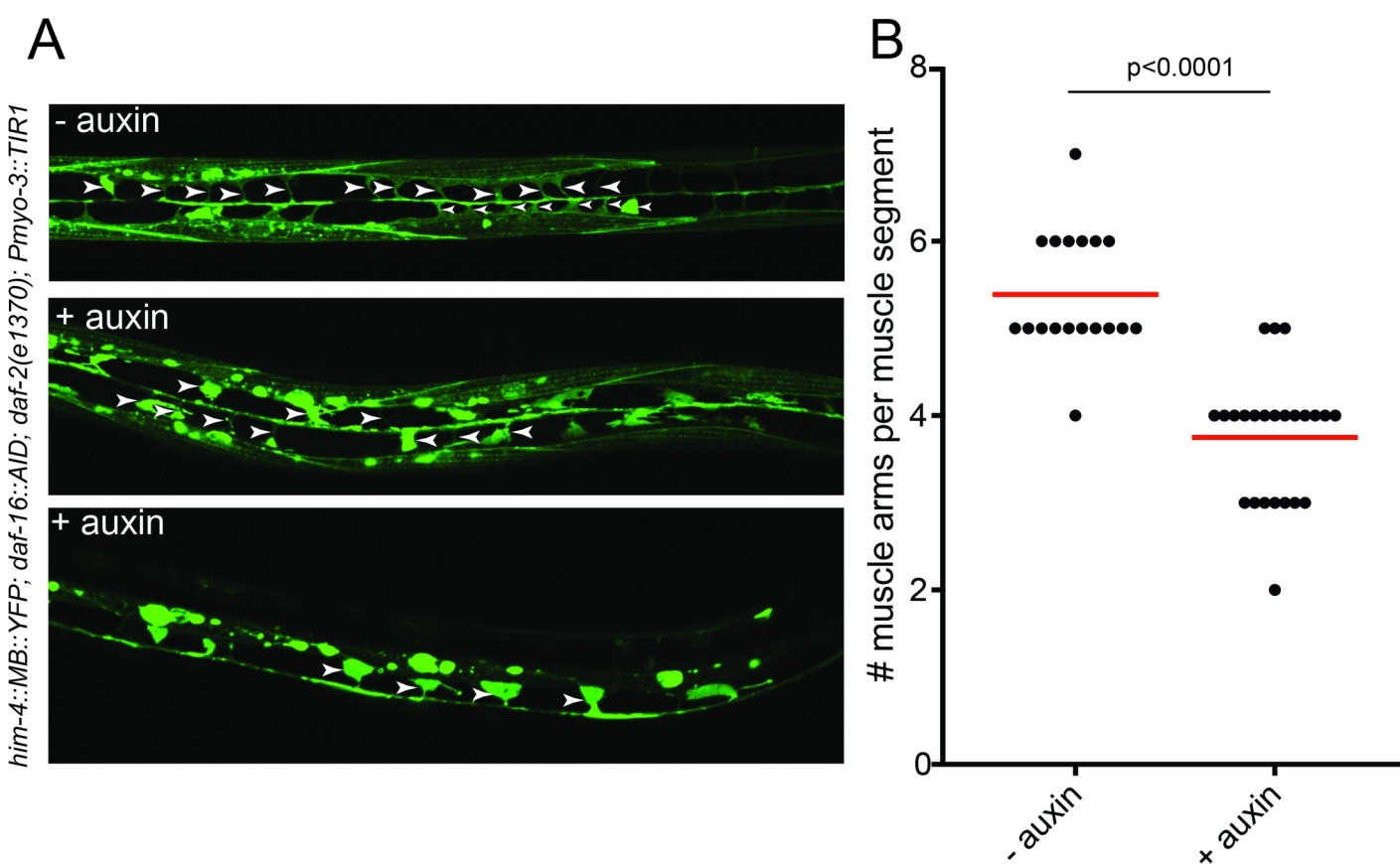

**Fig 11. DAF-16/FoxO acts cell-autonomously in muscle remodeling.** (**A**) Depleting DAF-16/FoxO from body wall muscle results in fewer muscle arms than in control dauers, as visualized with the *trIs30[him-4p::MB::YFP]* reporter [60]. (**B**) Quantification of the number of muscle arms per muscle segment in control and auxin-treated dauers. The data underlying this figure can be found in S1 Data. AID, auxin-inducible degron.

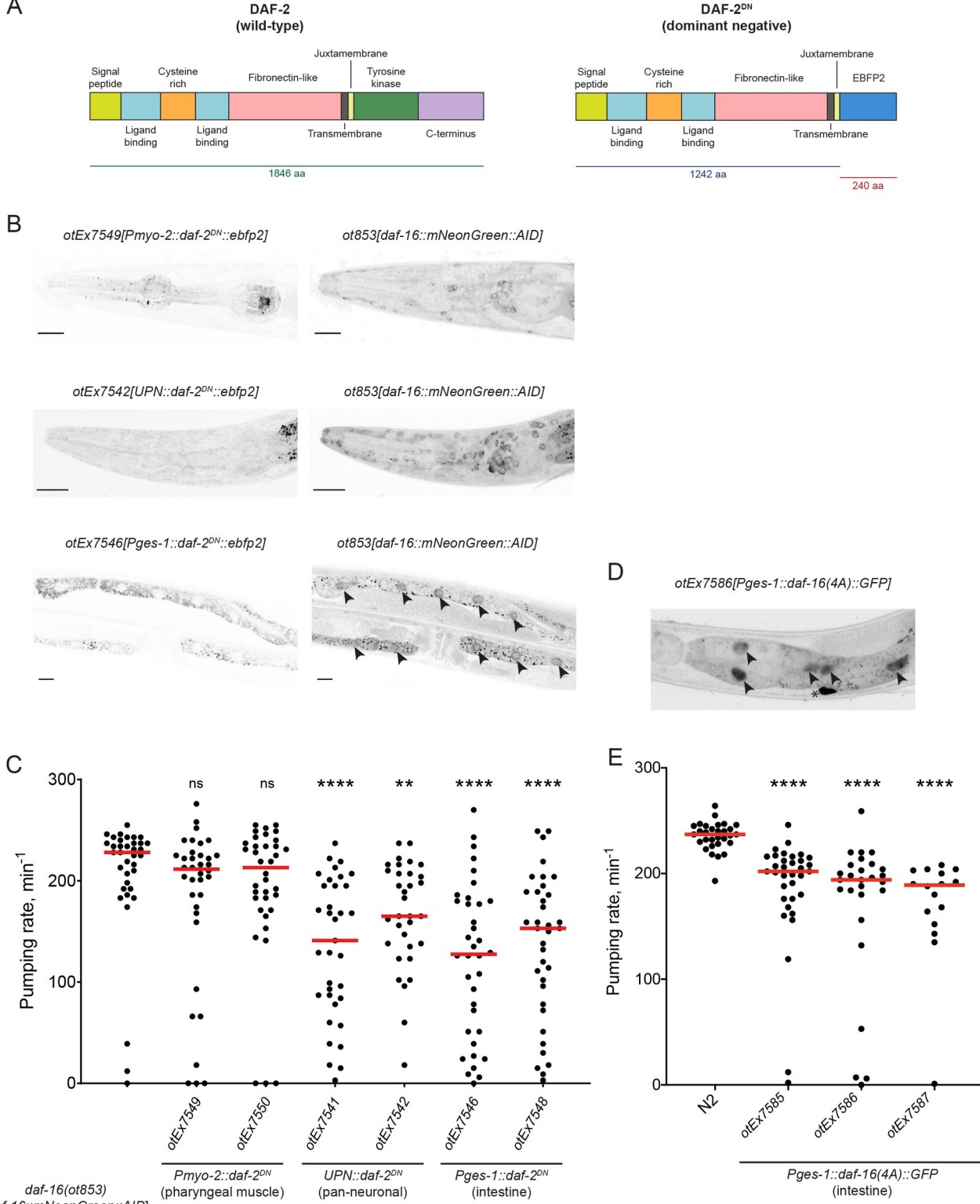

**Fig 12. DAF-16/FoxO activation via inhibition of insulin signaling is sufficient to cell nonautonomously reduce pharyngeal pumping. (A)** Schematic showing protein domains of wild-type DAF-2 (left) and DAF-2$^{DN}$, a dominant negative form of DAF-2/InsR in which the tyrosine kinase and carboxyl-terminal domains are replaced with the blue fluorescent protein EBFP2 (right). (**B**) Left panels: Expression of *daf-2$^{DN}$::ebfp2* in pharyngeal muscle (top), panneuronal (middle), and intestinal (bottom) tissues. Right panels: Localization of endogenously mNeonGreen-tagged DAF-16 protein

in the same animals shown in the corresponding panels on the left. Black arrowheads show nuclear localization of DAF-16 in intestinal tissue. Scale bars, 20 μm. (**C**) Pharyngeal pumping rate in day 1 adult animals expressing *daf-2^DN^::ebfp2* in pharyngeal muscle (*otEx7549* and *otEx7550*), panneuronal (*otEx7541* and *otEx7542*), and intestinal (*otEx7546* and *otEx7548*) tissues in the endogenously mNeonGreen-tagged *daf-16(ot853)* genetic background. Red horizontal lines represent median for ≥35 animals per condition. **, ****, and ns indicate $p < 0.01$, $p < 0.0001$, and $p ≥ 0.05$ compared to the *daf-16(ot853)* control strain in Dunn's multiple comparisons test performed after 1-way ANOVA on ranks. (**D**) Expression of GFP-tagged DAF-16(4A) in intestinal tissue of animals. Black arrowheads show strong nuclear localization of DAF-16(4A)::GFP. Asterisk indicates non-specific autofluorescence signal. (**E**) Pharyngeal pumping rate in day 1 adult wild-type (N2) strain and transgenic animals expressing *daf-16(4A)::gfp* in intestinal tissue (*otEx7585-87*). Red horizontal lines represent median for ≥16 animals per condition. **** indicates $p < 0.0001$ compared to wild-type strain in Dunn's multiple comparisons test performed after 1-way ANOVA on ranks. The data underlying this figure can be found in S1 Data. AID, auxin-inducible degron.

collaboration with terminal selector-type TFs, master regulatory TFs that initiate and maintain neuron type–specific gene expression programs during nervous system differentiation [64]. For example, DAF-16/FoxO-dependent up-regulation of a fosmid-based *che-7* reporter in the BDU, NSM, or AIM neurons may be the result of *daf-16* cooperating with the terminal selector of BDU, NSM, and AIM neuron identity, the *unc-86* POU homeobox gene (which acts in distinct cofactor combinations in these different neuron types) [65]. To test this hypothesis, we asked whether the dauer-specific up-regulation of *che-7* in BDU, NSM and AIM requires *unc-86*. Using *unc-86* null mutant animals, we indeed found this to be the case (**Fig 13A and 13B**). Similarly, the DAF-16/FoxO-dependent up-regulation of the GPCR *sri-9* in the NSM neuron also requires *unc-86* (**Fig 13C**). In conclusion, the cellular specificity of DAF-16/FoxO activity appears to be dictated by the neuron type–specific complement of terminal selectors.

## Discussion

The ability of cell types to undergo a change in phenotype ("cellular plasticity") is a hallmark of many different cell types throughout the animal kingdom. How such cellular plasticity is controlled on a single-cell level and how such plasticity is coordinated through many tissue types of an organism are fascinating questions that we address in this paper. The *C. elegans* dauer-remodeling paradigm highlights the importance of hormonal signaling events. Classic genetic epistasis analysis has revealed complex relationships between these hormonal signaling events [25], but many previous studies have focused on relatively crude binary readouts (whether animals go into dauer or not) and/or used experimental approaches that entailed certain limitations (most notably, problematic overexpression approaches that could have neomorphic effects; [57]). We have used CRISPR genome engineering technology to generate expression reagents and conditional alleles that allowed us to revisit a number of questions about the manner in which hormonal control defines cellular and behavioral remodeling events. Our genetic loss-of-function approaches have probed the genetic requirement of the key effector TFs of these hormonal signaling systems and are orthogonal to previous rescue approaches, which probed the sufficiency, rather than necessity, of these hormonal effector systems. Previous genetic mosaic analysis, particularly of the TGFβ [26] and insulin signaling systems [66], in the context of dauer formation, are conceptually similar to our tissue-specific protein depletion approach, but the latter has provided greater control over the cell types in which a specific protein is depleted, as well as about the timing of the action of these systems. We will first discuss timing and then discuss focus of action.

The temporal control of gene activity achieved through the AID system allowed us to address a previously unresolved question—are these hormonally controlled TF effectors of dauer remodeling only transiently required for entry into the dauer stage or is their activity continuously required? Since the vertebrate FoxO proteins have been proposed to act as pioneer TFs [67], a transient activity of at least DAF-16/FoxO appeared plausible, but our explicit demonstration of a continuous requirement of DAF-16/FoxO, as well as of DAF-3/SMAD and

 DAF-16 and DAF-12 control cellular plasticity both cell-autonomously and via interorgan signaling

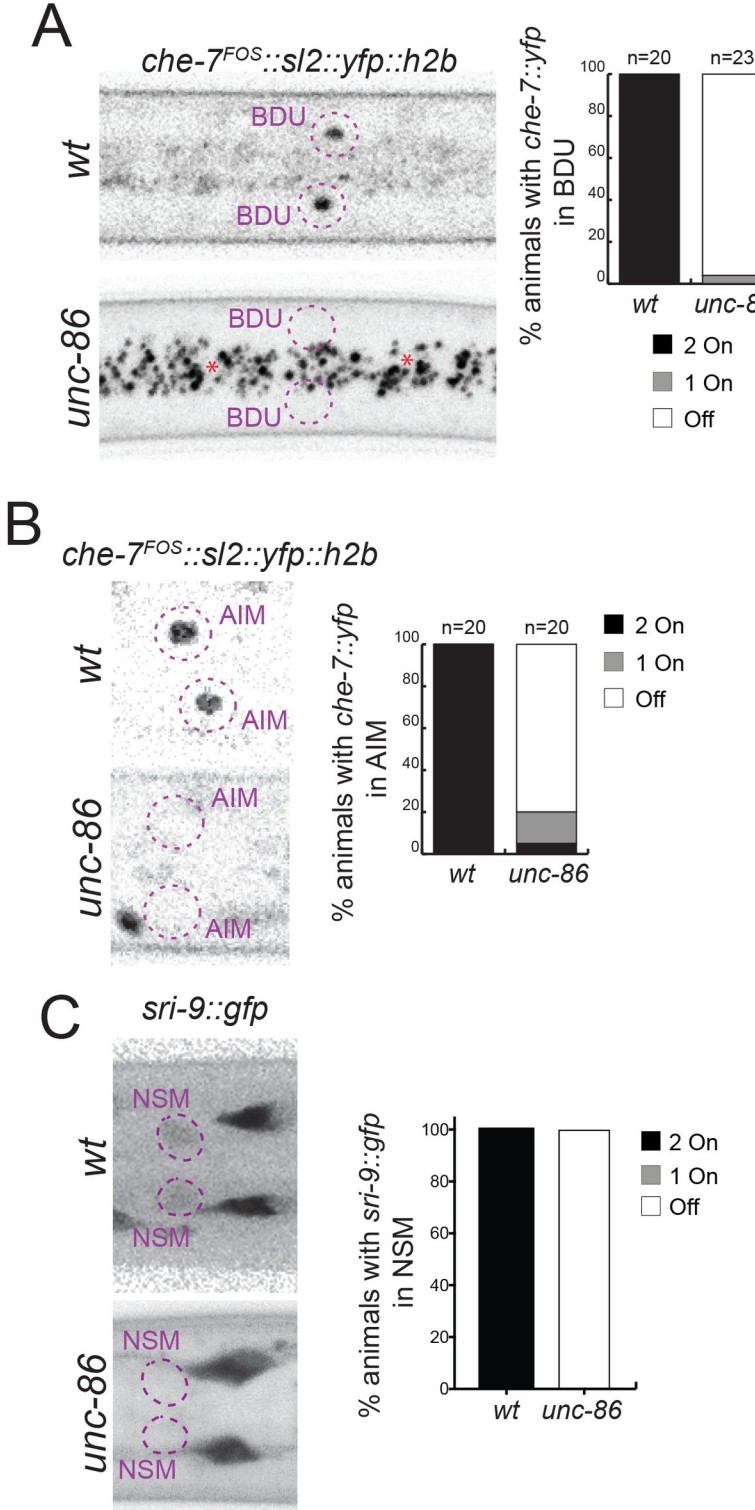

**Fig 13. Dauer-specific expression of neuronal reporters depends on terminal selector-type TFs.** (A) Dauer-specific expression of a *che-7* fosmid reporter (*otEx7112*) in BDU neurons is abolished in *unc-86(n846)* mutants. (B) Dauer-specific expression of a *che-7* fosmid reporter (*otEx7112*) in AIM neurons is abolished in *unc-86(n846)* mutants. (C) Dauer-specific expression of a *sri-9* reporter (*otIs732*) in NSM neurons is abolished in *unc-86(n846)* mutants. The data underlying this figure can be found in S1 Data. TF, transcription factor; wt, wild-type.

DAF-12/VDR, rather suggest that these TFs are likely continuously engaged on their target gene promoters to actively maintain the dauer state.

In regard to the focus of action of the 3 hormonal systems, our analysis (summarized in **Fig 14A**) has confirmed 2 key previously made conclusions about the focus of action of the TGFβ and insulin signaling systems: (1) TGFβ signaling via its effector TF DAF-3/SMAD operates within the nervous system to nonautonomously control tissue remodeling throughout the entire animal. We have extended this observation by demonstrating that TGFβ signaling is specifically required within sensory neurons, perhaps for the expression of insulin ligands that then signal to other cell types. (2) Insulin signaling via its effector TF DAF-16/FoxO operates cell-autonomously in a variety of distinct target tissues to control cellular remodeling. We have explicitly demonstrated such autonomy of DAF-16/FoxO function in a number of different tissue types, most extensively in the nervous system, where DAF-16/FoxO controls a plethora of instances of neuronal phenotypic plasticity. Our analysis of DAF-16/FoxO-depleted neurons was enabled by the unanticipated observation that neuronal DAF-16/FoxO depletion does not affect entry into the dauer stage. This observation was unanticipated because previous work has shown that resupplying *daf-16* in the nervous system rescues the Daf-c-suppression phenotype of *daf-16* mutants [28]. However, we cannot exclude the possibility that the AID system may not remove all protein function. Arguing against such possibility is that (a) in all cases examined, we do observe efficient tagged fluorescent protein removal; (b) AID-mediated DAF-16/FoxO depletion does phenocopy the null phenotype when ubiquitously implemented; and (c) AID-mediated neuronal DAF-16/FoxO depletion does result in a number of neuronal remodeling defects, indicating successful removal of DAF-16/FoxO. In conclusion, while the site of function for DAF-16/FoxO control of the initial entry into the dauer stage remains unresolved, our analysis clearly shows that DAF-16/FoxO operates cell-autonomously in the nervous system to execute neuronal remodeling.

In addition, our analysis revealed a plethora of unexpected cell nonautonomous activities of DAF-16/FoxO and DAF-12/VDR that indicates the existence of several interorgan signaling axes (summarized in **Fig 14A and 14B**). While neuronal DAF-16/FoxO may display a cell-autonomous requirement for pharyngeal pumping behavior, neuronal DAF-16/FoxO also affects the remodeling of pharyngeal muscle, as well as the metabolic remodeling of intestinal tissue observed upon dauer entry. Neuropeptidergic signaling axes from the nervous system to the gut have been described [68,69], and it is possible that during dauer remodeling, these axes become modulated in the nervous system by DAF-16/FoxO. How neuronal DAF-16/FoxO affects the autophagy-dependent constriction of pharyngeal musculature is presently entirely unclear, as is the manner by which intestinal DAF-16/FoxO affects pharyngeal muscle constriction and behavior. Notably, pharyngeal muscle constriction and cessation of pumping behavior are not obligatorily linked. Constricted pharynxes can be made to pump (**S7C Fig**) and, vice versa, pharyngeal pumping can cease without muscle constriction [70].

Another intriguing interorgan signaling axis that we discovered here is from the intestine to the pharyngeal nervous system, resulting in a down-regulation of pharyngeal pumping (**Fig 14B**). Our experiments indicate that DAF-16/FoxO is not only required, but also sufficient to instruct the intestine to control the activity of the pharyngeal nervous system. Since DAF-16/FoxO also acts within the pharyngeal nervous system, we hypothesize that intestinal DAF-16/FoxO modulates the release of an insulin signal from the intestine that controls DAF-16/FoxO activity in the pharyngeal neurons to control pumping behavior. Similarly, the effect of neuronal DAF-16/FoxO on intestinal DAF-16/FoxO-dependent metabolic remodeling suggests the DAF-16/FoxO-dependent release of insulin from the nervous system. The abundance and complex expression patterns of insulin-encoding genes in *C. elegans* [71,72] provide plenty of room for such multidirectional "insulin relay" systems (**Fig 14B**).

A

| Plasticity phenotype | Depletion | Effect of DAF-16 removal | Effect of DAF-12 removal |
|---|---|---|---|
| dauer entry | panneuronal | **NOT affected** | **NOT affected** |
| dauer entry | intestinal | **NOT affected** | **NOT affected** |
| dauer entry | body wall muscle | **NOT affected** | not tested |
| dauer entry | hypodermis | **NOT affected** | not tested |
| dauer entry | pharyngeal muscle | **NOT affected** | **NOT affected** |
| locomotory behavior | panneuronal | affected | affected* |
| locomotory behavior | body wall muscle | **NOT affected** | not tested |
| pharyngeal pumping | panneuronal | affected | affected |
| pharyngeal pumping | pharyngeal muscle | affected | **NOT affected** |
| pharyngeal pumping | intestinal | affected | **NOT affected** |
| pharyngeal pumping | pharyngeal neurons | affected | not tested |
| pharyngeal pumping | glutamatergic pharyngeal neurons | affected | not tested |
| pharyngeal pumping | cholinergic pharyngeal neurons | affected (weak effect) | not tested |
| pharyngeal muscle constriction | panneuronal | affected | affected |
| pharyngeal muscle constriction | pharyngeal muscle | affected | **NOT affected** |
| pharyngeal muscle constriction | intestinal | affected | **NOT affected** |
| muscle arms | body wall muscle | affected | not tested |
| *inx-6* induction in AIB | AIB interneurons | affected | affected |
| *sri-9* induction | panneuronal | affected in BDU, NSM, AIM, PHA, OLL | affected in NSM, OLL, AIZ, AWC |
| *sra-25* induction in ADL | panneuronal | **NOT affected** | not tested |
| *che-7* expression | panneuronal | affected in NSM, OLL, AIZ, ADA | not tested |
| *inx-2* downregulation | panneuronal | **NOT affected** | not tested |
| *gcy-5* induction in AWC$^{OFF}$ | AWC neurons | affected | not tested |
| fat staining | intestinal | affected | affected (not as strong) |
| fat staining | panneuronal | affected | affected |

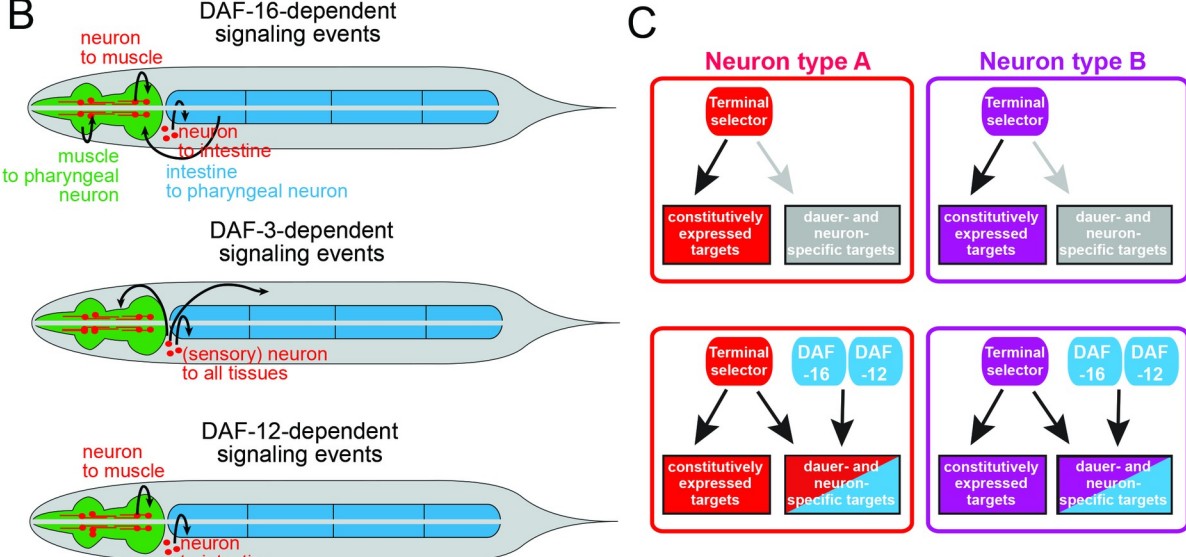

**Fig 14. Summary and conceptual models.** (**A**) Tabular summary of phenotypic outcomes of tissue-specific depletion of DAF-16/FoxO and DAF-12/VDR. (**B**) Models of inter-tissue signaling during dauer remodeling. (**C**) Terminal selectors control the expression of many immutable phenotypic features of a neuron, but are also enabling hormone-responsive TFs (DAF-12 and DAF-16) to induce dauer-specific target genes (e.g., *inx-6* in AIB or *gcy-6* in AWC). Future experiments will determine whether these interactions occur on the level of the *cis*-regulatory control elements of dauer specifically induced genes. One could envision that these control elements are tuned to not be responsive to either transcriptional input alone (terminal selector, dauer-specific TF), but requiring them both together. Since DAF-12 is commonly thought to act as a repressor in its unliganded form [25], these interactions may also be indirect. Not shown here is a variation of this model that may explain how DAF-16 may act to control the down-regulation of specific, dauer-repressed genes. Those could be under control of a terminal selector in non-dauer stage animal, but DAF-16 may antagonize its ability to activate these genes(s) upon entry into the dauer stage.

Lastly, our tissue-specific DAF-12/VDR removal experiments provide insights into the focus of DAF-12/VDR action in tissue remodeling and its relationship to DAF-16/FoxO. The genetic relationship of these 2 TF systems is complex and has mostly been studied in the context of the developmental arrest or organismal aging [45,53,73–75]. We have shown here a range of striking similarities in the effect of cell type–specific depletion of DAF-16/FoxO or DAF-12/VDR on the remodeling of distinct tissue types. Both TFs act autonomously in the nervous system to affect neuronal remodeling events and nonautonomously in the nervous system to bring about metabolic remodeling of the intestine. It is conceivable that they interact on the level of a target gene promoter to, for example, promote the expression of an insulin gene that may trigger intestinal remodeling. However, intestinal depletion of DAF-12/VDR, while also affecting intestinal remodeling, does not bring about pharyngeal remodeling. This indicates a distinct target gene spectrum of DAF-16/FoxO and DAF-12/VDR in the intestine, with only DAF-16/FoxO controlling the expression of a signal to the pharyngeal nervous system.

The mode of interaction between DAF-12/VDR and DAF-16/FoxO in the nervous system is presently unclear. While there are many phenotypic similarities between DAF-12/VDR and DAF-16/FoxO, the phenotypes of DAF-12/VDR generally appear more restricted. Even the effects on the same target gene shows cell type–specific differences in DAF-12/VDR and DAF-16/FoxO dependence. The up-regulation of the GPCR *sri-9* requires both DAF-12/VDR and DAF-16/FoxO in 4 neuron classes, but only DAF-16/FoxO in an additional 2 neuron classes. The different spectra of activity argue that DAF-16/FoxO and DAF-12/VDR do not obligatorily act together, but that these 2 TFs can have their own independent functions.

It is fascinating to consider how these plasticity phenomena intersect with the genetically hardwired differentiation programs of individual neuron classes, executed right after the birth of a neuron. These hardwired differentiation programs are controlled by terminal selector-type TFs [64]. One emerging theme appears to be that both hormonal systems intersect with the ability of terminal selectors to exert their functions (**Fig 14C**). For example, the dauer-specific up-regulation of a number of effector genes, such as the *inx-6* gene in AIB or the *che-7* gene in BDU, requires a cell type–specific terminal selector TF together with DAF-16/FoxO (UNC-42/Prop1 in AIB or UNC-86/Brn3 in BDU) [11, this paper]. In the case of *inx-6* and AIB, we find that not only *daf-16* but also *daf-12* is required for the *unc-42*-dependent up-regulation of *inx-6*. It is presently not clear whether this is an indication of DAF-12/VDR operating as a transcriptional activator or whether it operates indirectly via a repressive mechanism. In any case, the ability to intersect with distinct terminal selectors in distinct neuron classes may explain how globally expressed, hormone-responsive TFs like DAF-12 or DAF-16 can exert highly neuron type–specific effects (**Fig 14C**).

To what extent do the findings that we describe here relate to other organisms? There are intriguing molecular parallels between plasticity phenomena that control diapause in *C. elegans* and environmentally controlled developmental arrest and remodeling in other organisms. Environmentally controlled diapause in insects is also regulated by a steroid hormone, ecdysone, whose production relies on the ortholog of the same enzyme, DAF-36, as do the steroidal ligands (DAs) for DAF-12/VDR in *C. elegans* [76,77]. Moreover, the ecdysone nuclear hormone receptor EcR is also thought to control neuronal circuit remodeling in a cell-autonomous manner in *Drosophila*, albeit in the context of metamorphosis [78]. Insulin-like peptides have, like in *C. elegans*, also been implicated in controlling diapause in insects [79]. The insulin and steroid hormonal systems do not only operate in controlling environmentally controlled arrest in molting animals, but they also operate in vertebrates. Killifish, the only vertebrate known to undergo diapause, also employ vitamin D signaling and insulin signaling to control diapause [80]; however, the cellular remodeling events that accompany killifish diapause are not well explored. Killifish diapause arrest can occur right before hatching, after its entire

nervous system has formed [81], suggesting that involves neuronal remodeling events as well. More generally, the vertebrate DAF-12 homolog, vitamin D receptor (VDR), is, like DAF-12, broadly expressed in the brain and mediates the effects that vitamin D has on neuronal development and plasticity in mammals [32]. Similarly, insulin and insulin-like peptides have been shown to affect neuronal plasticity in a variety of different contexts in the mammalian brain [82,83]. In any of these systems, including *C. elegans*, it will be fascinating to explore through what effector genes these hormonal signals operate and how the cellular specificity of such effector systems is controlled.

Diapause and dispersal are usually viewed as alternative strategies to weather detrimental environmental conditions, with diapause reflecting an escape in time, and dispersal reflecting an escape in space [84]. In animals, diapause usually involves metabolic remodeling and an overall decrease in some aspects of behavior (e.g., locomotion) with the overarching goal of decreasing energy expenditure. On the other hand, the drive to disperse, an attempt to seek out more favorable conditions, requires more sophisticated changes in animal behavior, as evidenced, for example, in altered locomotion and in nictation behavior in nematodes. Intriguingly, the 2 alternative diapause and dispersal strategies are coupled in some animals, such as in the dauer stage (or the analogous infectious larval stage) of nematodes. Both diapause and dispersal demand that the gene regulatory state of a great variety of cells remains plastic, in order to permit the proper changes to the cellular phenotype, which, in turn, affect overall animal physiology and behavior. It is fascinating to appreciate that a hormonal effector like *daf-16/FoxO* (and to great extent also *daf-12/VDR*) appear to be employed across an enormously wide range of cells in very different tissue types to cell-autonomously trigger all these aspects of diapause and dispersal behavior. For example, *daf-16/FoxO* both acts in the process of metabolic remodeling, as well as in the nervous system to affect complex behaviors associated with dispersal strategies, ranging from sensory perception to locomotion. Understanding the mechanisms by which *daf-16/FoxO* (and *daf-12/VDR*) can exert such a great variety of effects represent a fascinating goal for future studies.

## Materials and methods

### Mutant and transgenic strains

The following mutant strains of *C. elegans* were used in this study:

CB1370 *daf-2(e1370)*
CB1372 *daf-7(e1372)*
MT1859 *unc-86(n846)*
CB1338 *mec-3(e1338)*
DR26 *daf-16(m26)*

The following transgenic strains of *C. elegans* were used in this study:

BC13401 *sIs12199 [sra-25::GFP]* [85]
OH3333 *otIs178 [flp-4::gfp]*
OH13405 *ot804 [inx-6::sl2::yfp::H2B]* [11]
OH13102 *otIs586 [gcy-6::NLS::GFP]* [86]
OH13908 *ot821 [daf-16::mKate2::3xFlag]* [87]
OH14125 *ot853 [daf-16::mNG::3xFlag::AID]* [11]
OH14654 *ot853 [daf-16::mNG::3xFlag::AID]; daf-2(e1370)*
OH14888 *ot853 [daf-16::mNG::3xFlag::AID]; daf-2(e1370); ieSi57 [Peft-3::TIR1::mRuby]*
OH14897 *ot853 [daf-16::mNG::3xFlag::AID]; daf-2(e1370); ieSi60 [Pmyo-2::TIR1::mRuby]*
OH14898 ot853 *[daf-16::mNG::3xFlag::AID]; daf-2(e1370); otTi1 [Pmyo-3::TIR1::mRuby]*
OH14945 *ot853 [daf-16::mNG::3xFlag::AID]; daf-2(e1370); ieSi61 [Pges-1::TIR1::mRuby]*

OH14915 *ot853 [daf-16::mNG::3xFlag::AID]; daf-2(e1370); otTi27 [Pdpy-7::TIR1::mTur2]*

OH15845 *ot853 [daf-16::mNG::3xFlag::AID]; daf-2(e1370); otIs730 [Punc-11prom8+ehs-1prom7+ rgef1prom2::TIR1::mTur2]*

OH16029 *ot975 [daf-16::mNeptune2.5::3xFlag::AID]* (Aghayeva et al., 2020)

OH14892 *ot875 [daf-3::GFP::3xFlag]*

OH14896 *ot877 [daf-3::TagRFP::3xFlag AID]*

OH14985 *ot877 [daf-3::TagRFP::3xFlag AID]; daf-7(e1372)*

OH14946 *ot877 [daf-3::TagRFP::3xFlag::AID]; daf-7(e1372); ieSi57 [Peft-3::TIR1::mRuby]*

OH14948 *ot877 [daf-3::TagRFP::3xFlag::AID]; daf-7(e1372); ieSi60 [Pmyo-2::TIR1::mRuby]*

OH15914 *ot877 [daf-3::TagRFP::3xFlag::AID]; daf-7(e1372); otIs730 [Punc-11prom8+ehs-1prom7+ rgef1prom2::TIR1::mTur2]*

OH14589 *ot870 [daf-12::GFP::3xFlag]*

OH14891 *ot874 [daf-12::TagRFP::3xFlag::AID]*

OH14984 *daf-7(e1372); ot874 [daf-12::TagRFP::3xFlag::AID]*

OH14986 *ot874 [daf-12::TagRFP::3xFlag::AID]; daf-7(e1372); ieSi57 [Peft-3::TIR1::mRuby]*

OH14988 *ot874 [daf-12::TagRFP::3xFlag::AID]; daf-7(e1372); ieSi61 [Pges-1::TIR1::mRuby]*

OH14989 *ot874 [daf-12::TagRFP::3xFlag::AID]; daf-7(e1372); ieSi60 [Pmyo-2::TIR1::mRuby]*

OH15913 *ot874 [daf-12::TagRFP::3xFlag::AID]; daf-7(e1372); otIs730 [Punc-11prom8+ehs-1prom7+ rgef1prom2::TIR1::mTur2]*

OH15597 *ot804 [inx-6::SL2::YFP::H2B]; ot853 [daf-16::mNG::3xFlag::AID]; daf-2(e1370); Ex [inx-1p::TIR1::mTur2::rps-27::NeoR +unc-122::GFP]*

OH16106 *sIs12199 [sra-25::GFP]; ot853 [daf-16::mNG::AID]; daf-2(e1370); otIs730 [Punc-11prom8+ehs-1prom7+rgef1prom2::TIR1::mTur2]*

OH16119 *ot853 [daf-16::mNG::3xFlag::AID]; daf-2(e1370); otIs730 [UPN::TIR1::mTur2]; otIs732 [Psri-9::GFP]*

OH16176 *daf-2(e1370); ot853 [daf-16::mNeonGreen::3xFlag::AID]; otIs586 [gcy-6::NLS::GFP]; otEx7434 [ceh-36_delASE::TIR1::mTur2]*

OH16496 *daf-7(e1372); ot877 [daf-3::TagRFP::3xFlag::AID]; Ex [Pift-20::TIR1::mRuby + Punc-122::GFP]*

OH16571 *otEx7585 [Pges-1::daf-16(4A)::GFP; unc-122p::mCherry]*

OH16572 *otEx7586 [Pges-1:: daf-16(4A)::GFP; unc-122p::mCherry]*

OH16573 *otEx7587 [Pges-1:: daf-16(4A)::GFP; unc-122p::mCherry]*

OH16440 *ot853 [daf-16::mNG::3xFlag::AID]; otEx7541 [UPN::daf-2$^{DN}$::EBFP2; unc-122p::mCherry]*

OH16441 *ot853 [daf-16::mNG::3xFlag::AID]; otEx7542 [UPN:: daf-2$^{DN}$:: EBFP2; unc-122p::mCherry]*

OH16447 *ot853 [daf-16::mNG::3xFlag::AID]; otEx7546 [Pges-1:: daf-2$^{DN}$:: EBFP2; unc-122p::mCherry]*

OH16449 *ot853 [daf-16::mNG::3xFlag::AID]; otEx7548 [Pges-1:: daf-2$^{DN}$:: EBFP2; unc-122p::mCherry]*

OH16450 *ot853 [daf-16::mNG::3xFlag::AID]; otEx7549 [Pmyo-2::daf-2 daf-2$^{DN}$:: EBFP2; unc-122p::mCherry]*

OH16451 *ot853 [daf-16::mNG::3xFlag::AID; otEx7550 [Pmyo-2:: daf-2$^{DN}$:: EBFP2; unc-122p::mCherry]*

OH16665 *ot853 [daf-16::mNG::3xFlag::AID]; daf-2(e1370); otEx7628 [ehs-1prom4::TIR1::mTur2, unc-122p::mCherry]*

OH16666 *ot853 [daf-16::mNG::3xFlag::AID]; daf-2(e1370); otEx7629 [ehs-1prom4::TIR1::mTur2, unc-122p::mCherry]*

OH16752 *ot853 [daf-16::mNG::3xFlag::AID]); daf-2(e1370); otEx7668 [unc-17prom4::TIR1::mTur2, unc-122p::mCherry]*

OH16754 *ot853 [daf-16::mNG::3xFlag::AID]); daf-2(e1370); otEx7669 [unc-17prom4::TIR1::mTur2, unc-122p::mCherry]*

OH16766 *ot853 [daf-16::mNG::3xFlag::AID]); daf-2(e1370); otEx7670 [eat-4prom7::TIR1::mTur2, unc-122p::mCherry]*

OH16767 *ot853 [daf-16::mNG::3xFlag::AID]); daf-2(e1370); otEx7671 [eat-4prom7::TIR1::mTur2, unc-122p::mCherry]*

OH16694 *ot853 [daf-16::mNG::AID]; daf-2(e1370); otTi1 [Pmyo-3::TIR1::mRuby]; trIs30 [him-4p::MB::YFP + hmr-1b::DsRed2 + unc-129nsp::DsRed2]*

OH16896 *ot975 [daf-16::mNeptune2.5::3xFlag::AID]; daf-2(e1370); otIs730 [UPN::TIR1::mTur2]; inx-2(ot906 [inx-2::sl2::yfp::h2b])*

OH16897 *ot975 [daf-16::mNeptune2.5::3xFlag::AID]; daf-2(e1370); otIs730 [UPN::TIR1::mTur2]; otEx7112 [che-7_fosmid::sl2::yfp::h2b; pha-1(+); myo-2::bfp]*

## Generation of tissue-specific TIR1 lines

TIR1 cassette was amplified from pLZ31[pCFJ350modified-*Peft3*-TIR1-linker-mRuby- *unc-54* 3′UTR] [41] and inserted into the miniMos plasmid pCFJ910 [88] carrying the neomycin resistance gene, NeoR, as a selectable marker. Tissue- or cell-specific promoters [UPN, *Pceh-36prom2* (ASE-), *Pmyo-3*, *Pdpy-7*, *Pinx-1*, *ehs-1prom4*, *eat-4prom7*, *unc-17prom4*] were inserted in the resulting miniMos plasmid using restriction-free (RF) cloning or restriction/ligation. AWC-specific promoter *ceh-36prom2* is an 1,852-bp (−1,883 to −32) fragment of the *ceh-36* promoter with a mutated ASE motif [37]. The UPN promoter was developed by Eviatar Yemini [40] by concatenating 3 promoter fragments of panneuronally expressed genes: *unc-11prom8*, *ehs-1prom7*, and *rgef-1prom2* [39]. *ehs-1prom4* is a 62-bp (−332 to −271) fragment of the *ehs-1* promoter that is expressed in all pharyngeal neurons (weak expression in I1 and I4) [39]. *eat-4prom7* is a 587-bp (−5,038 to −4,452) fragment of the *eat-4* promoter that is expressed in glutamatergic pharyngeal neurons I5 and M3 (weak expression in M5 and MI) [38]. *unc-17prom4* is an 837-bp (−822 to +15) fragment of the *unc-17* promoter that is expressed in cholinergic pharyngeal neurons I1, M1, M2, M4, M5, and MC (weak expression in I3; also expressed in AIY, IL2, RIH, and RIR) [38].

For the *UPN::TIR1*, *AWC::TIR1*, *Pdpy-7::TIR1*, *ehs-1prom4*, *eat-4prom7*, and *unc-17prom4* constructs, the fluorophore tagging TIR1 in the original plasmid (pLZ31) was replaced with mTurquoise2 that was amplified from pDD315 (mTurquoise2^SEC^2xHA), a gift from Daniel Dickinson (Addgene plasmid #73343 http://n2t.net/addgene:73343; RRID:Addgene_73343).

It should be noted that while single-copy TIR1 lines for nonneuronal tissues have resulted in specific protein depletion and quantifiable and specific phenotypes, we found that single-copy neuronal drivers of TIR1 were insufficient in that regard. Therefore, we used integrated or extrachromosomal multicopy arrays of TIR1 for neuronal expression (panneuronal or neuron type specific, i.e., AIB, AWC, and pharyngeal neuron specific).

## Conditional alleles of *daf-16*, *daf-3*, and *daf-12*

AID was amplified from pLZ29[pCFJ350modified-*Peft3*-AID(71–114)EmGFP-*unc-54*-3′UTR] [41] and inserted into pDD268 and pDD284 [89]. These plasmids were used for SEC-mediated CRISPR insertion of FP::AID tags into genomic loci of *daf-16*, *daf-12*, and *daf-3* [89].

## Generation of *daf-2^DN^* and *daf-16(4A)* constructs

To create *daf-2^DN^*, a dominant negative version of the insulin/IGF-like receptor *daf-2*, the cDNA for the *daf-2 a* isoform was modified using RF cloning to replace the sequence encoding

the tyrosine kinase and carboxyl-terminal domains (corresponding to amino acids 1,246 to 1,846) with the blue fluorescent protein *ebfp2* sequence. This *daf-2*[DN] sequence was subsequently cloned downstream of tissue-specific promoters (*Pmyo-2*, UPN, and *Pges-1*) and upstream of *unc-54* 3′UTR using RF cloning.

For creating an insulin signaling–independent form of *daf-16*, the cDNA for the *c* isoform of *daf-16* (referred to as the *daf-16a* in [36]) was synthesized de novo, and 4 Ser/Thr residues were replaced with alanine (T54A, S240A, T242A, and S314A). This *daf-16(4A)* sequence was subsequently translationally fused upstream of the *gfp* sequence of pPD95.75 plasmid, and the entire construct was cloned downstream of the intestine-specific promoter *Pges-1* using RF cloning.

## Auxin treatment

Auxin treatment was performed by transferring worms to bacteria-seeded plates containing auxin. The natural auxin indole-3-acetic acid (IAA) was purchased from Alfa Aesar (Ward Hill, MA) (#A10556). A 400 mM stock solution in ethanol was prepared and stored at 4° C for up to 1 month. Nematode Growth Medium (NGM) agar plates with fully grown OP50 bacterial lawn were coated with the auxin stock solution to a final concentration of 4 mM and used on the same day. All plates in the experiments were protected from light because of the light sensitivity of auxin.

## Oil Red O staining

Intestinal lipid staining with Oil Red O was performed according to a protocol by [90].

## Life span analysis

Life span assays for strains with tissue-specific DAF-16/FoxO depletion were performed using the multi-well WorMotel platform [91]. We prepared 2 to 4 independent replicates of each experiment, with total $N = 40$ to 80 worms per condition. More specifically, the following replicates were tested for each genotype: (1) *daf- 2(e1370)*: 3 independent experiments, $N = 20$ per condition; (2) *daf-2(e1370); daf-1::AID*: 4 independent experiments, $n = 20$ per condition; (3) *daf-2(e1370); daf-16::AID; Peft-3::TIR1*: 4 independent experiments, $n = 20$ per condition; (4) *daf-2(e1370); daf-1::AID; Pmyo-2:TIR1*: 2 independent experiments, $n = 20$ per condition; (5) *daf-2(e1370); daf-16::AID; Pmyo-3:TIR1*: 2 independent experiments, $n = 20$ per condition; and (6) *daf-2(e1370); daf-16::AID; Pges-1:TIR1*: 2 independent experiments, $n = 20$ per condition.

Images of worms in the WorMotel were acquired using static imaging systems [92] with illumination from a blue LED array applied twice per day to stimulate movement. Survival curves were calculated as previously described [91].

## Microscopy

Worms were anesthetized using 100 mM sodium azide (NaN$_3$) and mounted on 5% agarose pads on glass slides. Images were acquired as Z-stacks of approximately 0.7 μm-thick slices with Zen software (ZEN Digital Imaging for Light Microscopy, RRID:SCR_013672) using a Zeiss LSM700 confocal microscope (Jena, Germany). Images were reconstructed via maximum intensity Z-projection of 2- to 0-μm Z-stacks using Zen software.

## Automated worm tracking experiments

Automated single worm tracking was performed using WormTracker 2.0 system [93] at 25°C. Animals were recorded for 5 minutes to ensure sufficient sampling of locomotion-related

behavioral features. Dauer and non-dauer animals were placed on uncoated NGM plates that were kept at 25˚C before recording. To minimize any potential bias arising due to dauer stage–specific prolonged bouts of spontaneous pausing [17], we only selected animals that were actively moving at the beginning of the assay. To avoid potential variability arising due to fluctuations in room conditions, all strains that were compared in a single experiment were recorded on the same day.

### Pharyngeal pumping rate assays

Constitutive dauer animals on an NGM plate containing OP50 bacteria were observed under 50× objective lens of a Nikon (Tokyo, Japan) Eclipse E400 upright microscope equipped with DIC optics. The number of pharyngeal pumps in a 2-minute period was measured for at least 15 animals per condition. For the N2 and *daf-16(m26)* strains that were not dauer constitutive, dauer animals were isolated from a starved plate after treatment with 1% SDS for 30 minutes. SDS-resistant dauer individuals were picked and transferred to a fresh NGM plate seeded with a uniform thin layer of OP50 bacteria and allowed to acclimate for 15 minutes. Subsequently, pharyngeal pumps were scored (as described above) within 40 minutes of placing the dauer animals on the OP50 lawn plate.

For measurement of pharyngeal pumping rate in adults, day 1 adult worms grown at 25˚C from hatching were picked and transferred from an uncrowded non-starved plate to a fresh NGM plate seeded with a uniform thin layer of OP50 bacteria. Worms were allowed to recover for 10 minutes and were subsequently visualized under 20× objective lens of a Nikon Eclipse E400 upright microscope. The number of pharyngeal pumps in a 1-minute period was measured for at least 15 animals per condition. All pharyngeal pumping assays were performed at room temperature.

## Supporting information

**S1 Fig. GFP images indicating successful removal of TFs after auxin treatment.** (**A**) Confocal images of dauers with panneuronal depletion of DAF-16/FoxO under control and auxin treatment conditions. Scale bar, 20 μm. (**B**) Confocal images of dauers with intestinal depletion of DAF-16/FoxO under control and auxin treatment conditions. Insets show an example of an intestinal nucleus, magnified and enhanced to boost fluorescence signal (both in control and auxin-treated conditions). Scale bar, 20 μm. (**C**) Epifluorescent images of dauers with body wall muscle depletion of DAF-16/FoxO under control and auxin treatment conditions. Insets show an example of a muscle nucleus, magnified and enhanced to boost fluorescence signal. Scale bar, 20 μm. (**D**) Confocal images of dauers with panneuronal depletion of DAF-12/VDR under control and auxin treatment conditions. Pharyngeal expression is from a co-injection marker *inx-6prom 18::TagRFP*. Scale bar, 20 μm. TF, transcription factor.
(TIF)

**S2 Fig. DAF-3::GFP expression in starved larvae.** Expression of the *daf-3::GFP* CRISPR allele at different stages in development in starved conditions. Anterior is to the left on all images. Scale bar, 20 μm (same for all images).
(TIF)

**S3 Fig. Quantification of SDS resistance of dauers of the conditional strains.** Worms were washed off NGM plates (control and auxin-treated) 3 days after rearing at 25˚C and incubated in 1% (m/v) solution of SDS for 30 minutes, with continuous gentle shaking. After washing with water and M9 buffer, the worms were plated on fresh plates and scored as alive if moving. The data underlying this figure can be found in S1 Data. NGM, Nematode Growth Medium.
(TIF)

**S4 Fig. Intestinal rescue of daf-16 results in incomplete rescue of the Daf-c phenotype of daf-2; daf-16 mutants. (A)** DIC images of a *daf-2(e1370)* dauer and a dauer-like larva with intestinal rescue of *daf-16b/d/f* isoforms in the *daf-2(e1370); daf-16(Df50)* background. Note the deviation from the wild-type filariform pharynx morphology in the latter case. **(B)** Quantification of pharyngeal activity of the strains depicted in (A). The data underlying this figure can be found in S1 Data.
(TIF)

**S5 Fig. Dauer-specific expression change of electrical synapse components is independent of intestinal DAF-16/FoxO activity. (A)** A *che-7* reporter *(otEx7112)* expression is gained in NSM neurons in dauer. This dauer-specific *che-7* expression is lost upon panneuronal depletion of DAF-16/FoxO in auxin-treated dauers. **(B)** Dauer-specific *che-7* expression in NSM is unaffected upon intestinal depletion of DAF-16/FoxO in auxin-treated dauers. **(C)** Expression of an *inx-6* reporter allele *(ot804)* is gained in AIB neurons in dauer. This dauer-specific *inx-6* expression in AIB neurons is unaffected upon intestinal depletion of daf-16/FoxO in auxin-treated dauers. **(D)** Expression of an *inx-2* reporter allele *(ot906)* is down-regulated in multiple neurons in dauer. This dauer-specific down-regulation of *inx-6* expression is unaffected upon intestinal depletion of DAF-16/FoxO in auxin-treated dauers. The data underlying this figure can be found in S1 Data.
(TIF)

**S6 Fig. (A)** Locomotory features of dauers with DAF-16 depleted from neurons and body wall muscle and DAF-12 depleted from neurons, as compared to their respective vehicle controls. **(B)** Locomotory features of dauers with DAF-16 depleted from neurons and body wall muscle, as compared to *daf-2(e1370)* control dauers. **(C)** Effect of DAF-16 depletion from body wall muscle on dauer morphology. The data underlying this figure can be found in S2 Data.
(TIF)

**S7 Fig. Pharyngeal constriction and pharyngeal pumping can be decoupled. (A)** Pharyngeal morphology in wild-type (N2) and *daf-16(m26)* starvation-induced dauers. Yellow arrows show width of the terminal bulb of the pharynx. Scale bars, 20 μm. **(B)** Pharyngeal pumping rate in N2 and *daf-16(m26)* starvation-induced dauer animals. Blue horizontal line represents median for ≥25 animals per genotype. ns indicates $p = 0.36$ in 2-tailed Mann–Whitney test. **(C)** Pharyngeal pumping rate in wild-type animals recovering from starvation-induced dauer stage. The data underlying this figure can be found in S1 Data.
(TIF)

**S1 Data. Collection of all primary numerical data shown in Fig 3B, 3D; Fig 4A–4C; Fig 5A–5D; Fig 6A–6E; Fig 8A, 8C–8D; Fig 9B, 9D, 9F, 9H, 9I; Fig 10B, 10C; Fig 11B; Fig 12C, 12E; Fig 13A–13C; S3 Fig; S4B Fig; S5A–S5D Fig; S7B–S7C Fig.**
(XLSX)

**S2 Data. Collection of all primary numerical data shown in Fig 7A–7B and S6A–S6C.**
(XLSX)

## Acknowledgments

We thank Chi Chen for generating transgenic strains, Tulsi Patel for originally noting the *gcy-6* expression change in dauers, and Piali Sengupta and Michael O'Donnell for comments on the manuscript.

## Author Contributions

**Conceptualization:** Ulkar Aghayeva, Oliver Hobert.

**Formal analysis:** Ulkar Aghayeva, Abhishek Bhattacharya, Surojit Sural, Matthew Churgin.

**Investigation:** Ulkar Aghayeva, Abhishek Bhattacharya, Surojit Sural, Eliza Jaeger, Matthew Churgin.

**Project administration:** Oliver Hobert.

**Supervision:** Christopher Fang-Yen, Oliver Hobert.

**Writing – original draft:** Oliver Hobert.

**Writing – review & editing:** Ulkar Aghayeva, Oliver Hobert.

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
