## [Editor Report · Decision Letter 0]

17 Dec 2020

Dear Dr Hobert, 

Thank you for submitting your manuscript entitled "DAF-16/FoxO and DAF-12/VDR control cellular plasticity both cell-autonomously and via interorgan signaling" for consideration as a Research Article by PLOS Biology.

Your manuscript has now been evaluated by the PLOS Biology editorial staff as well as by an academic editor with relevant expertise and I am writing to let you know that we would like to send your submission out for external peer review.

Please re-submit your manuscript within two working days, i.e. by Dec 21 2020 11:59PM.

I want to let you know that we will not be able to send your manuscript to reviewers until after the Holidays, as the PLOS Biology office will be closed for part of the next two weeks and we have found that reviewer acceptance rates are typically very low over the Holiday season. I will start inviting reviewers first thing in the new year.

Given the disruptions resulting from the ongoing COVID-19 pandemic, there may also be additional delays in the editorial process. We apologize in advance for any inconvenience caused and will do our best to minimize impact as far as possible.

Kind regards,

Lucas Smith, Ph.D.,

Associate Editor

PLOS Biology

---

## [Decision Letter · Decision Letter 1]

3 Feb 2021

Dear Dr Hobert,

Thank you very much for submitting your manuscript "DAF-16/FoxO and DAF-12/VDR control cellular plasticity both cell-autonomously and via interorgan signaling" for consideration as a Research Article at PLOS Biology. Your manuscript has been evaluated by the PLOS Biology editors, an Academic Editor with relevant expertise, and by several independent reviewers.

You will see that the reviewers are all very positive and think that the findings are important and significant for the field. Nevertheless, they also have some comments that need to be addressed.

In light of the reviews (below), we are pleased to offer you the opportunity to address the points from the reviewers in a revised version that we anticipate should not take you very long. We will then assess your revised manuscript and your response to the reviewers' comments and we may consult the reviewers again.

We expect to receive your revised manuscript within 1 month.

**IMPORTANT: In your revision, please also make sure to address the data and other policy-related requests noted below.

**IMPORTANT - SUBMITTING YOUR REVISION**

*Resubmission Checklist*

Please make sure to read the following important policies and guidelines while preparing your revision (and please address the specific requests included below my signature):

*Published Peer Review*

*PLOS Data Policy*

*Blot and Gel Data Policy*

Sincerely,

Lucas Smith, Ph.D.,

Associate Editor,

lsmith@plos.org,

PLOS Biology

SPECIFIC DATA POLICY REQUESTS: 

Fig 3B,D; Fig 4A-C; Fig 5A-D; Fig 6A-E; Fig 7A-B; Fig 8 A,C,D; Figure 9B,D,F,H,I; Fig 10B,C; Fig 11B; Fig 12C,E; Fig 13 A,B,C; Fig S3; Fig S4B; Fig S5 A-D; Fig. S6 A-C; Fig S7 B,C

BLURB REQUEST:

When resubmitting, please also provide a blurb which (if accepted) will be included in our weekly and monthly Electronic Table of Contents, sent out to readers of PLOS Biology, and may be used to promote your article in social media. The blurb should be about 30-40 words long and is subject to editorial changes. It should, without exaggeration, entice people to read your manuscript. It should not be redundant with the title and should not contain acronyms or abbreviations. For examples, view our author guidelines: https://journals.plos.org/plosbiology/s/revising-your-manuscript#loc-blurb

REVIEWS:

Reviewer's Responses to Questions

PLOS authors have the option to publish the peer review history of their article (what does this mean?). If published, this will include your full peer review and any attached files.

Reviewer #1: Yes: Patrick Hu

Reviewer #2: No

Reviewer #3: No

Reviewer #4: No

Reviewer #1: The C. elegans dauer larva is an excellent model for studying the molecular basis for phenotypic plasticity. Molecular pathways that govern dauer formation are evolutionarily conserved, suggesting that mechanisms underlying phenotypic plasticity in C. elegans dauers will inform studies on phenotypic plasticity in other organisms. While the logic of insulin-like, TGFbeta-like, and hormonal signaling pathways that control dauer entry and commitment is reasonably well understood, how the three major transcription factor targets of these pathways function to execute the dauer developmental fate is largely unknown. In this manuscript the authors use state-of-the-art reagents they have developed to perform a detailed analysis of the spatiotemporal requirements for dauer-specific phenotypic plasticity. The study is enabled by the generation of strains that are tagged at endogenous transcription factor loci with fluorescent proteins and an auxin-inducible degron (AID). These strains permit spatiotemporal control of gene expression while obviating caveats of previous studies using transgenic constructs that overexpress genes and may lack regulatory elements that control endogenous gene expression.

The experiments presented define sites and times of action for DAF-3/SMAD, DAF-16/FoxO, and DAF-12/VDR in the execution of various aspects of the dauer developmental program. An enormous amount of data is presented. The experiments are well-executed, and the data are convincing. Taken together, the data provide new insights into temporal, cell-autonomous, and cell-nonautonomous actions of DAF-16/FoxO and DAF-12/VDR in dauer execution, confirm previous studies on the nonautonomy of TGFbeta and insulin-like signaling in dauer regulation, and resolve conflicting data in the literature regarding sites of action of DAF-16/FoxO in dauer regulation. The paper will be of great interest and utility to dauer mavens as well as developmental biologists, cancer biologists, and anyone with an interest in phenotypic plasticity.

I have a few minor comments:

1. Fig. 12B: in light of the unexpected result that DAF-2DN expression in the pharynx does not induce nuclear translocation of DAF-16 in the pharynx, it would be nice to know the extent to which ubiquitous overexpression of DAF-2DN induces dauer arrest in wild type animals at 25C. My guess is that even in the context of DAF-2DN overexpression, a low level of wild-type DAF-2 dimers form and induce DAF-16 phosphorylation and cytoplasmic retention. This is not absolutely necessary but could be included if the authors have this data in hand.

2. Typo in Fig. S2: "intestinal"

3. Fig. S4: "mgDf50"

Reviewer #2: The manuscript by Hobert and colleagues investigates longstanding questions about the site of action and timing of regulatory pathways involved in the initiation and maintenance of the dauer pathway of C. elegans. Dauer formation is a remarkable instance of developmental plasticity in nematodes, but in addition, the genetic analysis of dauer entry of C. elegans has served as a paradigm for understanding the role of evolutionarily conserved neuroendocrine signaling pathways in organismal physiology. This important study addresses longstanding questions in the field through the incisive application of CRISPR-based-methods that represent a marked improvement over prior analyses, and the findings define how cell-non-autonomous signaling can coordinate organismal responses across tissues.

The generation and analysis of strains carrying in-genome tags to monitor expression revealed DAF-3 expression to be increased with starvation. DAF-16 is shown to undergo nuclear localization in the intestine, but not in the neurons upon food deprivation, revising and adding to prior studies of DAF-16 localization in response to stress, but here with a more rigorous and solid experimental foundation. DAF-12 expression is also analyzed. Further engineering of auxin-inducible degron tags, coupled with tissue-specific expression of the TIR1-interaction domain, enable the authors to conduct essentially targeted mosaic-like analysis in a cell-type specific manner with temporal control. DAF-3 is noted to function in the nervous system, and somewhat surprisingly in the ciliated neurons given the prior report of Ashrafi and colleagues (Greer et al., 2008) reporting evidence that DAF-1 functions in RIM/RIC interneurons. This may merit some discussion. The DAF-16 discussion is delicately handled but provides very interesting and again, more solidly grounded (in terms of methodology, there are fewer caveats) data that will be extremely helpful for the many investigators studying DAF-16 in diverse aspects of C. elegans physiology. The authors carefully note that tagging of the genes involved do not grossly affect function in the dauer entry phenotypes/suppression are left unaffected, but the reduced lifespan of daf-2(e1370);daf-16::AID relative to daf-2 (Fig 10) that is noted by the authors may merit some further discussion, as these data suggest some degree of reduction of function of DAF-16 from the tagging. Importantly, I do not think this caveat affects the major conclusions from the experiments using daf-16::AID in the manuscript, as these experiments are all internally controlled, and if anything, the effects observed are likely to be relatively smaller if the tagging affects baseline activity. The authors are also able to simply and elegantly define the roles of these pathways in not only initiation, but maintenance of the dauer state, though the differing kinetics of recovery may merit some discussion. For this reviewer, these data would be plenty enough for a highly impactful study.

But the authors go on to use these strains to define, in rigorous and elegant detail, all of the findings that are basically summarized in the table in Figure 14. Far from simply being a listing of phenotypes that are modified by cell-autonomous vs. cell-non-autonomous activities, these data begin to put together a picture of organismal physiology around dauer initiation, entry, and maintenance, decoupling some aspects specific to individual pathways from the overall phenotype, and underscoring the manner in which neuronal signaling can control organismal physiology. There are a few points that I would suggest might benefit from further discussion, and Figure 2A has a typo ("instestinal nuclei" next to the purple circles), but nothing that takes away substantively from this beautifully conceived, rigorously executed, and clearly presented story. 

Reviewer #3: This is a masterful and complete genetic characterization of how the global transcriptional output of insulin, TGF beta, and the daf-12 NHR ligands remodel a growing reproducing animal into the very different dauer animal of C. elegans. The experiments are complex but very well explained, and the data presentation is very clear. It is written with a focus on neural activity that subserves the changes from dauer to non dauer life cycles. And it is written in a manner that allows even non experts in C. elegans to follow the logic. The literature on where insulin signals for dauer arrest vs reproductive development has been conflicting because of transgene issues and this paper clears it all up. This is a large field in C. elegans and so there will be many interested readers. But my one suggestion to the authors is to add a paragraph (am I kidding? This paper is already super long) about how the circuits of neural response to insulin signaling defects are likely to present in the millions of insect species that undergo diapause and try to disperse to new niches. Also, dauers are the dispersal stage of C. elegans, hitching a ride onto insects to fly to new ecosystems when the current niche begins to suck. Dispersal stages of insects probably use ancestrally related pathways, including the neurons that this paper highlights. This paper does not have prove that the circuits are present in insects but they can discuss the possibility. But the paper needs no additional experiments. Read my lips Reviewers 2 and 3: No additional experiments. There are a ton of great experiments in the many figures. Very impressive. 

But many of the Figures can become Supplementary Figures

Here are the Figures that I suggest become Supplementary

Figure 1

Figure 3AC

Figure 4 fuse to Figure 3 BD

Figure 7

Figure 8

Figure 10

Figure 12

I would not suggest these moves except that a 14 Main Figure paper is longer than any I have ever seen. The self assurance of the authors is impressive. 

 Congratulations to the authors on producing such a wonderful paper.

Reviewer #4: The manuscript describes a set of new tools to assess the temporal and tissue-specific requirement of daf-3, daf-16, daf-12, three major transcription factors in regulating dauer formation in C. elegans. In general, the study is well-designed and well executed. The data presented are rigorous and the conclusions drawn are appropriate. 

The manuscript is fairly complex to get through, because a lot of details were being presented. Figure 14 certainly helps, but perhaps additional summary figure can be added in some of the earlier figures. e.g. the various consequences of neuronal depletion of daf-16 and daf-12 can be compared and summarized earlier. Then another summary figure can be used for the pharyngeal phenotypes. Further, the idea presented in Figure 13 deserves an additional model figure to illustrate the bigger picture model about collaboration between DAF-16 and terminal selector TFs. 

Also, regarding to overall dauer decision, the data seem to suggest that elimination of DAF-16 in a single tissue is not sufficient to block the dauer program. It follows then that whether the authors have tried removal of daf-12 or daf-16 in more than one tissue? e.g. neuron combined with intestine, etc?

---

## [Editor Report · Decision Letter 2]

16 Mar 2021

Dear Dr Hobert,

Thank you for submitting your revised Research Article entitled "DAF-16/FoxO and DAF-12/VDR control cellular plasticity both cell-autonomously and via interorgan signaling" for publication in PLOS Biology. I have now obtained advice from the Academic Editor.

We think that the study is impactful and well executed, and that your revision has adequately addressed the reviewer concerns. However, before we can formally accept your manuscript, I need you to address two minor editorial issues, outlined here and included again below my signature.

1) Please provide a blurb, which (if accepted) will be included in our weekly and monthly Electronic Table of Contents, sent out to readers of PLOS Biology, and may be used to promote your article in social media. The blurb should be about 30-40 words long and is subject to editorial changes. It should, without exaggeration, entice people to read your manuscript. It should not be redundant with the title and should not contain acronyms or abbreviations. For examples, view our author guidelines: https://journals.plos.org/plosbiology/s/revising-your-manuscript#loc-blurb

2) Thank you very much for providing a supplementary file, containing the underlying data for each of your figures (S1_data). Would please add a sentence to each of the figure legends (including supplementary figure legends) stating where the underlying data can be found? For example, you could add the sentence: “The data underlying the figure can be found in S1 Data.”

We expect to receive your revised manuscript within two weeks. 

*Published Peer Review History*

*Early Version*

Sincerely,

Lucas Smith, Ph.D.,

Associate Editor,

lsmith@plos.org,

PLOS Biology

DATA POLICY:

Please also ensure that figure legends in your manuscript include information on where the underlying data can be found.

BLURB

Please provide a blurb, which (if accepted) will be included in our weekly and monthly Electronic Table of Contents, sent out to readers of PLOS Biology, and may be used to promote your article in social media. The blurb should be about 30-40 words long and is subject to editorial changes. It should, without exaggeration, entice people to read your manuscript. It should not be redundant with the title and should not contain acronyms or abbreviations. For examples, view our author guidelines: https://journals.plos.org/plosbiology/s/revising-your-manuscript#loc-blurb

---

## [Editor Report · Decision Letter 3]

23 Mar 2021

Dear Dr Hobert,

On behalf of my colleagues and the Academic Editor, Jennifer Garrison, I am pleased to say that we can in principle offer to publish your Research Article "DAF-16/FoxO and DAF-12/VDR control cellular plasticity both cell-autonomously and via interorgan signaling" in PLOS Biology, provided you address any remaining formatting and reporting issues. These will be detailed in an email that will follow this letter and that you will usually receive within 2-3 business days, during which time no action is required from you. Please note that we will not be able to formally accept your manuscript and schedule it for publication until you have made the required changes.

PRESS

Thank you again for supporting Open Access publishing. We look forward to publishing your paper in PLOS Biology. 

Sincerely, 

Lucas Smith, Ph.D. 

Senior Editor 

PLOS Biology